# Computational modeling and quantitative physiology reveal central parameters for brassinosteroid-regulated early cell physiological processes linked to elongation growth of the *Arabidopsis* root

Ruth Großeholz[1,2]*[†], Friederike Wanke[3][†], Leander Rohr[3], Nina Glöckner[3], Luiselotte Rausch[3], Stefan Scholl[1], Emanuele Scacchi[3,4], Amelie-Jette Spazierer[3], Lana Shabala[5], Sergey Shabala[5,6], Karin Schumacher[1], Ursula Kummer[1,2][‡], Klaus Harter[3]*[‡]

[1]Centre for Organismal Studies, Heidelberg University, Heidelberg, Germany; [2]BioQuant, Heidelberg University, Heidelberg, Germany; [3]Center for Molecular Biology of Plants, University of Tubingen, Tübingen, Germany; [4]Department of Ecological and biological Science, Tuscia University, Viterbo, Italy; [5]Tasmanian Institute for Agriculture, University of Tasmania, Hobart, Australia; [6]International Research Centre for Environmental Membrane Biology, Foshan University, Foshan, China

*For correspondence:
ruth.grosseholz@bioquant.uni-heidelberg.de (RG);
klaus.harter@zmbp.uni-tuebingen.de (KH)

[†]These authors contributed equally to this work
[‡]These authors also contributed equally to this work

Competing interest: The authors declare that no competing interests exist.

**Abstract** Brassinosteroids (BR) are key hormonal regulators of plant development. However, whereas the individual components of BR perception and signaling are well characterized experimentally, the question of how they can act and whether they are sufficient to carry out the critical function of cellular elongation remains open. Here, we combined computational modeling with quantitative cell physiology to understand the dynamics of the plasma membrane (PM)-localized BR response pathway during the initiation of cellular responses in the epidermis of the *Arabidopsis* root tip that are be linked to cell elongation. The model, consisting of ordinary differential equations, comprises the BR-induced hyperpolarization of the PM, the acidification of the apoplast and subsequent cell wall swelling. We demonstrate that the competence of the root epidermal cells for the BR response predominantly depends on the amount and activity of H⁺-ATPases in the PM. The model further predicts that an influx of cations is required to compensate for the shift of positive charges caused by the apoplastic acidification. A potassium channel was subsequently identified and experimentally characterized, fulfilling this function. Thus, we established the landscape of components and parameters for physiological processes potentially linked to cell elongation, a central process in plant development.

## Editor's evaluation

The manuscript by Grosseholz et al. addresses a so far much overlooked aspect in plant hormone signalling systems, namely how components primarily identified by genetic means jointly regulate plant root physiology. The authors demonstrate a relevant effect of Brassinosteroid signalling on the control of root cell elongation in Arabidopsis. They used an elegant combination of (electro)

physiology, computer modelling and confocal microscopy to reveal molecular candidates linking BR perception to fine-tuning of cell elongation through ion fluxes.

## Introduction

Brassinosteroids (BRs) are plant steroid hormones that regulate a great variety of physiological and developmental processes including elongation growth as well as environmental adaptations (*Müssig et al., 2002*; *Clouse, 2002*; *Lv and Li, 2020*; *Wolf, 2020*). To achieve this, BR signal transduction is closely linked with a multitude of other signaling pathways (*Lv and Li, 2020*).

The canonical sequence of BR perception and signal transduction, which also leads to cell elongation, is mediated by the plasma membrane (PM)-resident, nanoscale-organized receptor kinase brassinosteroid-insensitive 1 (BRI1) and its co-receptor BRI1-activating kinase 1 (BAK1) as central elements (*Bücherl et al., 2013*; *Bücherl et al., 2017*; *Lv and Li, 2020*; *Wolf, 2020*). The binding of BR to the receptor's extracellular domain results in the re-arrangement of several BRI1-associated proteins. This involves the release of inhibitory mechanisms that include BRI1 kinase inhibitor 1 (BKI1) and BAK1-interacting receptor like kinase 3 (BIR3) and leads to the stabilization of BRI1/BAK1 association followed by a variety of auto- and trans-phosphorylation events of their cytoplasmic domains. This cascade of events eventually results in the establishment of the fully active BRI1 receptor complex (*Bücherl et al., 2013*; *Bücherl et al., 2017*).

Once the active complex is established, the BR response is proposed to divide into two distinct downstream pathways to trigger molecular and physiological processes, which can be linked to cell elongation and differ in their kinetic properties (*Clouse, 2002*; *Clouse, 2011*; *Vukašinović et al., 2021*): A long-term (hours to days) gene regulatory pathway leading to extensive transcriptional reprogramming that is realized *via* the kinase Brassinosteroid Insensitive 2 (BIN2), the key transcription factors brassinazole resistant 1 (BZR1) and BR insensitive EMS suppressor 1 (BES1). The gene regulatory pathway is linked to cell wall remodeling as well as the extent and correct timing of anisotropic cell growth (*Lv and Li, 2020*; *Fridman et al., 2021*; *Graeff et al., 2021*). Physiological work in the past already suggested the second, short-term pathway is proposed to occur in PM-resident, nano-organized BRI1 complexes (*Cerana et al., 1983*; *Cerana et al., 1984*; *Romani et al., 1983*; *Mandava, 1988*). The response takes place in a matter of a few minutes and leads to the upregulation of the major proton pumping ATPases (AHA1, AHA2) (*Figure 1*; *Elgass et al., 2009*; *Caesar et al., 2011*). The activation of AHAs involves their interaction with BRI1 and BAK1, is BRI1 kinase activity-dependent (*Caesar et al., 2011*; *Ladwig et al., 2015*) and occurs very likely via rapid phosphorylation (within 5 min) of threonine and serine residues in the AHAs' large cytoplasmic domain (*Lin et al., 2015*; *Witthöft et al., 2011*). The BR-enhanced activity of AHAs induces the acidification of the apoplastic space, hyperpolarization of the PM's membrane potential ($E_m$) and cell wall swelling within 10 min after BR application (*Elgass et al., 2009*; *Caesar et al., 2011*; *Witthöft et al., 2011*; *Witthöft and Harter, 2011*). The functional link between these BR-regulated cellular responses and AHA activity was proven by the inhibition or constitutive activation of the pump, leading either to the blocking of the reactions or their activation in the absence of BR (*Caesar et al., 2011*).

The cell wall swelling, thus the incorporation of water in the wall matrix, is mediated by the loosening of the walls rigidity *via* the activation of acidic pH-dependent, apoplast-resident enzymes regulating wall extensibility (*Cosgrove, 2000*). According to the acid-growth theory (*Hager, 2003*), the low pH-induced enzymatic loosening of the cell wall, often paralleled by the accumulation of osmotically active substances inside the cell, causes a water potential difference between the extracellular space and the protoplast, the uptake of water and eventually the onset of cell elongation (*Palmgren et al., 2009*; *Regenberg et al., 1995*; *Baekgaard et al., 2005*; *Phyo et al., 2019*). This sequence of short- and long-term signaling and reaction pathways allows for instance root cells in the elongation zone (EZ) to grow four times their size in the meristematic zone (MZ) with a growth rate of up to 0.7 µm min$^{-1}$(*Fasano et al., 2001*; *Verbelen et al., 2006*). Comparable growth rates were reported for hypocotyl cells of dark-grown *Arabidopsis* seedlings upon application of BR (*Minami et al., 2019*).

While the activation of the pathway is well understood qualitatively, the information on the inactivation of the pathway is currently still sparse. The receptor BRI1 autophosphorylates at the residue S891, which inhibits the receptor activity (*Oh et al., 2012*). However, the time-scale of this phosphorylation is very slow, as it increases over the course of 12 h after stimulation with BR. The dephosphorylation

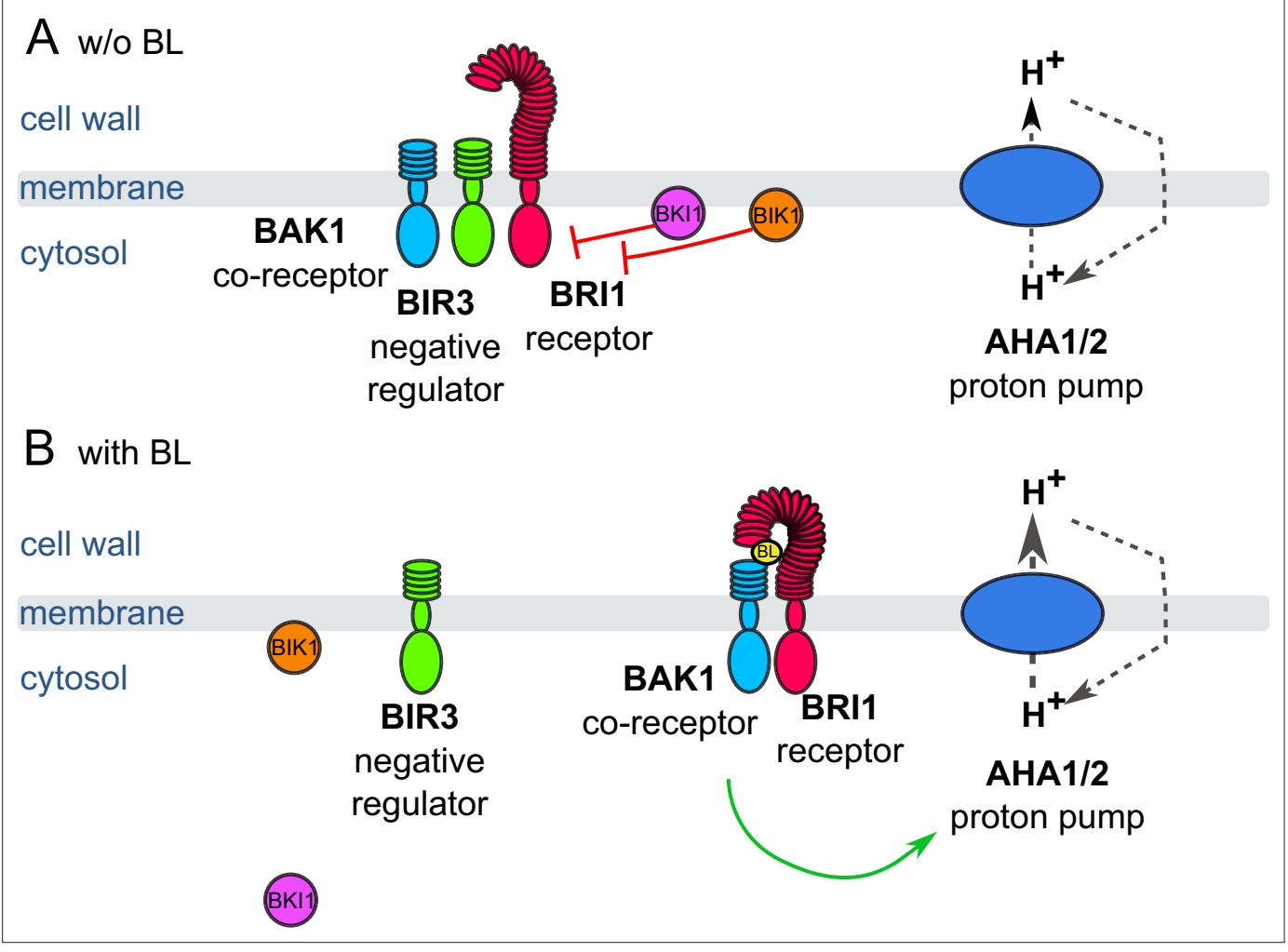

**Figure 1.** Schematic overview of the key constituents and processes of the plasma membrane-associated fast BR response pathway initiating early steps in cell elongation, here represented by brassinolide (BL). (**A**) Inactive state: Co-localizing in a preformed nano-orgnaized complex, the inhibitors BKI1, BIK1 and BIR3 suppresses the activity of BRI1 in the absence of BL keeping the activity of H⁺-ATPases AHA1 and 2 at basic levels. By interaction with BAK1, BIR3 blocks the access of the co-receptor to BRI1. (**B**) Active state: Upon BL-binding to the receptor, the inhibitory mechanisms of BKI1, BIK1 and BIR3 on BRI1 and BAK1 are released causing the formation of the active BRI1/BAK1 complex. The complex enhances the AHA activity resulting in cell wall acidification, plasma membrane hyperpolarization and eventually onset of cell elongation. These key constituents and qualitatively described processes were used for the initial establishment of the computational model at cellular.

of this site is even slower, as residual phosphorylations can be detected 5 days after inhibiting BR synthesis using brassinazole (*Oh et al., 2012*).

Despite the qualitative knowledge on the constituents, the BR perception and the canonical signaling events, the dynamics of the system as a whole have yet to be examined quantitatively in detail (*Sankar et al., 2011*; *van Esse et al., 2013a*; *van Esse et al., 2012*; *van Esse et al., 2013b*; *Allen and Ptashnyk, 2017*). Therefore, we employed computational modeling in combination with quantitative experimental data on the fast BR response pathway in the PM, focusing on the epidermal cells of the *Arabidopsis* root tip as the epidermis limits the rate of elongation (*Hacham et al., 2011*). The root tip is an excellent model system for such a combined study because cells there first undergo a phase of cell division in the MZ followed by a phase of growth in the EZ. The boundary from the MZ to the EZ is represented by the transition zone (TZ). The formation of the TZ is characterized by the cytokinin-induced expression of the *AHA1* and *AHA2* genes as a precondition for cell elongation in the EZ (*Pacifici et al., 2018*). However, BR is involved in the control of both cell division and cell elongation in the different zones, apparently also adding to the specific functional competence and behavior of the cells along the axis of the root tip.

However, the molecular determinants and processes establishing this competence and their link to the cytokinin-caused gradient of growth competence are poorly understood in terms of their quantitative dynamics. This lack of knowledge virtually provokes the implementation of computational modeling.

While computational modeling has been used frequently in biomedical research since the early 2000 s, its application to the plant field has started more recently (*Hübner et al., 2011*; *Holzheu and Kummer, 2020*). Here, the growth and development of the root tip has been of particular interest (*Bruex et al., 2012*; *Muraro et al., 2016*; *Di Mambro et al., 2017*; *Rutten and Ten Tusscher, 2019*; *Salvi et al., 2020*; *Rutten and Ten Tusscher, 2021*). Further computational studies in plants include the modeling of auxin signaling (*Vernoux et al., 2011*) and transport pattern (*Band et al., 2014*), and parts of the BR signaling (*Sankar et al., 2011*; *van Esse et al., 2013a*; *van Esse et al., 2012*; *van Esse et al., 2013b*; *Allen and Ptashnyk, 2017*). For instance, the modeling approach by van Esse et al. analyzed the link between the BR dose, gene expression and growth behavior in both the *Arabidopsis* root and shoot (*van Esse et al., 2013a*; *van Esse et al., 2012*; *van Esse et al., 2013b*). However, none of the previous modeling approaches has been able to truly quantitatively depict cellular responses, make clear predictions about the cellular behavior, limiting constituents or processes.

In our study, we were able to determine how the constituents of the PM-resident fast BR response pathway work together and identified its rate-limiting elements applying an ordinary differential equations (ODE) approach. Substantiated by wet lab experiments, our computational approach led to a detailed kinetic model that describes the rapid cellular response and offers an explanation for the initiation of BR controlled differential growth behavior of the root cells on the basis of the differential AHA accumulation and activity. Furthermore, the model predicts the existence of a cation influx across the PM that is crucial for the apoplastic acidification and $E_m$ hyperpolarization, which was subsequently narrowed down experimentally. Lastly, the model shows how the extent of the BR response can be fine-tuned by the level of the BIR3 inhibitor. Our model proposes that the specific composition of the PM-resident nano-organized BRI1 complexes determines the competence of the root cells to initiate elongation in response to BR.

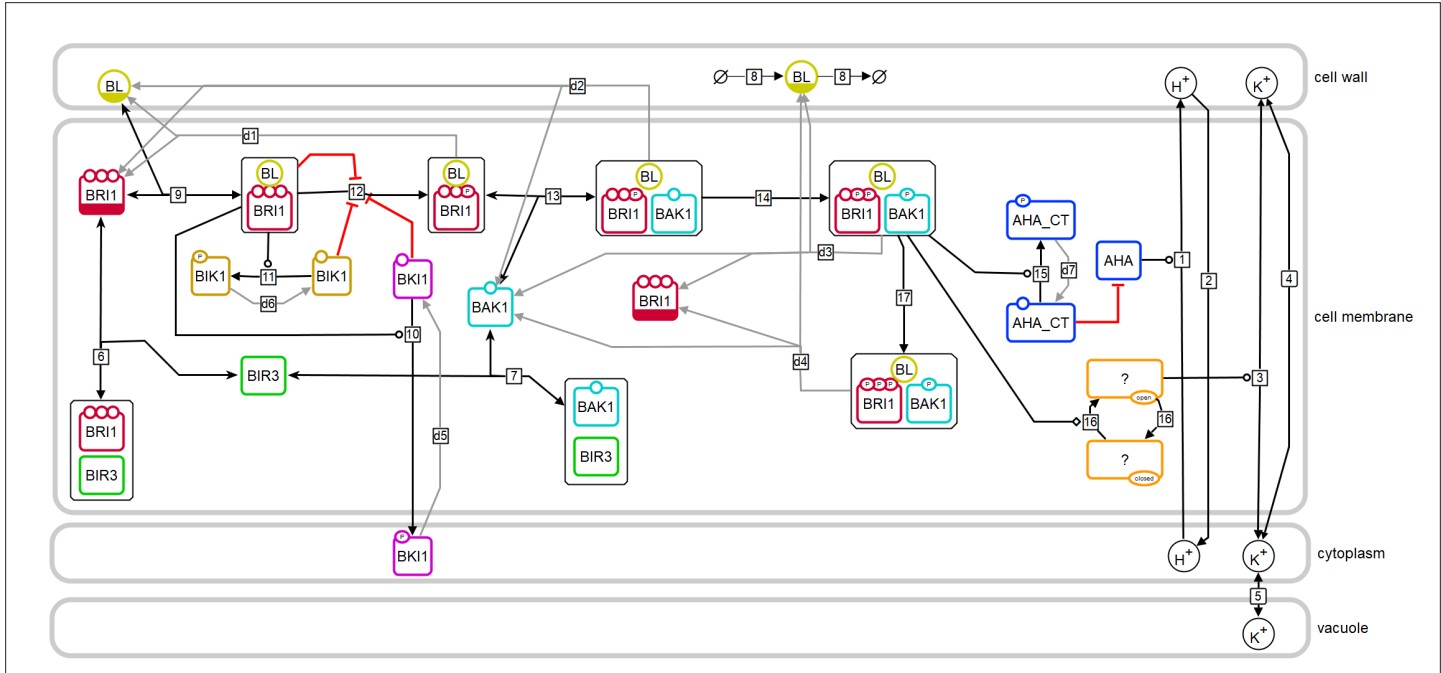

**Figure 2.** Model structure of the fast BR response pathway of *Arabidopsis thaliana*. Compartments are indicated by grey boxes. Smaller molecules are indicated by circles, proteins by rectangles. Potential sites for protein modifications are indicated by the small circles on the boundaries of the rectangles. Reactions, including substrates and products, are indicated by the arrows, with the reaction numbers noted in the small box. Reactions, which are required for the model to return to the initial state, are drawn in grey. A bar at the bottom of the circle or rectangle indicates that this entity appears more than once in the scheme.

# Results

## A mathematical model of the fast BR response

To analyze the important steps and factors of the cell-specific, fast BR response in the root tip, we developed a detailed mathematical model consisting of ODEs (*Figure 2*). The model comprises four cell compartments: the cytosol, the cell wall and the vacuole as three-dimensional compartments as well as the PM as a two-dimensional compartment. The explicit inclusion of the PM as two-dimensional compartment was prompted by the fact that most components of the BR perception and initial processes are located in the membrane and the relevance of the membrane as a scaling factor in this kind of system (*Holzheu et al., 2021*). The compartment sizes were set such that the model initially describes the behavior of a single epidermis cell in the early EZ of the *Arabidopsis* root (*Wilma van Esse et al., 2011*) (see *Appendix 1—table 1*).

The model captures the important components and steps of the fast BR response pathway focusing on protein interactions and post-translational modifications. We decided against the inclusion of data, which are derived from the genetic manipulation of component amounts or activity or are based on long-term incubation of BR biosynthesis inhibitors such as brassinazole. This kind of manipulation or long-term treatment are expected to have considerable effects on the physiological and developmental properties of the plant as a whole. The model is set up in a way that an equilibrium state was reached before the system is stimulated with the hormone by maintaining the system first without the hormone for 24 hr. In this state, only a few crucial reactions occur and carry a flux ($v$): the interaction between BIR3 and BAK1 ($v_7$) and BIR3 and BRI1 ($v_6$), the proton leak from the cell wall into the cytoplasm ($v_2$) (*Appendix 1—figure 1*), the basal activity of the ATPases AHA1 and AHA2 ($\nu_1$) and the exchange of monovalent cations (here represented by potassium) between cytoplasm and cell wall ($v_4$) and cytoplasm and vacuole ($v_5$). Modeling the basal state as a physiologically plausible steady state ensures that the model describes the inactive state of the BR response pathway accurately and that the interactions of BIR3 with BAK1 and BRI1 are in an equilibrium.

The hormone is added to the model by an event triggered at 24 h. According to the current state of knowledge, this initiates a number of molecular processes in the PM that occur almost simultaneously (*Figure 2*): binding of BL to BRI1 ($v_9$), the loss of BRI1 inhibition by its C-terminus ($v_{12}$), the release of BKI1 and BIK1 after phosphorylation ($v_{10}$ and $v_{11}$, respectively) as well as the release of BIR3 from BAK1, the establishment of the BAK1-BRI1 interaction *via* BR ($v_{13}$), and the auto- and transphosphorylation of BAK1 and BRI1 ($v_{14}$). These spatial rearrangements and post-translational modifications result in the active form of the BRI1 receptor complex, which immediately stimulates the activity of H⁺-ATPases very likely by phosphorylation (*Lin et al., 2015*) ($v_{15}$). Further signaling events occur later in time and include differential gene expression (*Lv and Li, 2020*). However, these late events were not considered here for our modeling approach.

The main cell physiological output of this early sequence of events is the acidification of the apoplastic space, the hyperpolarization of the $E_m$ and the swelling of the cell wall. The $E_m$ is calculated based on the net change in charge distribution of protons and potassium across the PM, the specific capacitance of the plasma membrane (*White et al., 1999*) and the membrane surface (*Wilma van Esse et al., 2011*) (see *Appendix 1—table 1*). However, combining the available information derived from the literature resulted in preliminary model draft that was not able to reproduce the measured experimental data, for instance regarding the $E_m$ hyperpolarization. Without a mechanism to balance out the shift in charge distribution caused by the transported protons, even a modest acidification of the apoplast from a pH of 5.4–5.0 will result in a non-physiological membrane hyperpolarization (*Sondergaard et al., 2004*) (see: Appendix 1 - example calculation of $E_m$ and pH change). Consequently, we postulated a cation influx in the model (here represented by potassium) that starts upon activation of the BRI1 complex ($v_{16}$) and is driven by the $E_m$ ($v_3$).

However, in order to accurately model and simulate the fast BR response pathway, we needed more experimental data about the PM-based BRI1 response module. Any remaining unknown model parameters were estimated based on experimental data of the cell wall acidification (this study), $E_m$ hyperpolarization (*Caesar et al., 2011*) and the qualitative overexpression behavior of BIR3 (*Imkampe et al., 2017*). To account for non-identifiable parameters, we investigated the parameter space by computing several independent model parameterizations that describe the experimental data equally well. All computational analyses were run with each model of the resulting ensemble of structurally identical models (n=10) to ascertain consistent results across parameter space.

## Quantification of signaling components

One experimental challenge for the refinement of the model was to quantify the central components of the pathway comprising predominantly BRI1, BAK1, BIR3 and AHA in the PM of epidermal cells of the root tip. Initially, we drew our attention on their steady-state transcript levels as they were determined by high-throughput single cell RNA-sequencing (scRNA-Seq) of the different *Arabidopsis* root cell types (*Ma et al., 2020*). Whereas *BRI1* and *BIR3* transcripts accumulated in all cell types of the root more or less equally and did not alter much in their amount during cell development along the root axis, *AHA2* and to lesser extent also *AHA1* transcripts were found predominantly in the epidermal cells and the root cortex (*Figure 3A*). During root development, the *AHA2* transcript amount but not those of *BRI1* and *BIR3* started to increase strongly in the cortex and epidermis cells of the TZ and EZ (*Figure 3B*). This temporal transcript pattern was less prominent for *AHA1* (*Figure 3B*) being in agreement with earlier observation that the *AHA1* promoter is not very active in root epidermis cells. This indicates that *AHA1* does not play a prominent role in the control of cell expansion (*Merlot et al., 2007*). Because its transcript accumulation was already induced by protoplasting, the scRNA-Seq data could not be used for *BAK1* with respect to the temporal expression along the root axis (*Ma et al., 2020*).

On the basis of the scRNA-Seq data we focused our further studies on the in vivo protein quantification of the GFP fusions of BRI1, BAK1, BIR3 and AHA2 in developing epidermal cells along the root tip axis. For the PM of cells of the EZ, the amount of BRI1-GFP was already quantified to around 11 receptor molecules per $\mu m^2$ and for BAK1-GFP to 5 co-receptors per $\mu m^2$ by *Wilma van Esse et al., 2011*. To complete this data set, we applied quantitative CLSM for the quantification of BIR3-GFP and AHA2-GFP in the epidermal root cells of published transgenic *Arabidopsis* lines that express the fusion protein under the respective native promoter (*Fuglsang et al., 2014*; *Imkampe et al., 2017*). As these GFP fusion proteins carry the identical fluorophore version, their fluorescence intensity can be set in relation to the BRI1-GFP intensity and, thus, to the BRI1-GFP receptor amount in the PM. The quantification of GFP fluorescence was performed in 50x50 μm areas at the epidermis along the root tip (an exemplary set of root tip images is shown in *Appendix 1—figure 2*). The amount of BRI1-GFP and BAK1-GFP did not alter much in the epidermal cells in the MZ and early EZ, as it was reported before (*Figure 3C*; *Wilma van Esse et al., 2011*). A relative homogeneous fluorescence intensity distribution was also observed for BIR3-GFP that translated to about 17 inhibitor molecules per $\mu m^2$ PM area in the MZ and 14 in the early EZ (*Figure 3C*). In contrast, there was a significant gradient of AHA2-GFP fluorescence intensity along the root axis, being comparatively low in the MZ (with 4 AHA2 molecules per $\mu m^2$ PM area) but high in the late EZ / maturation zone (with about 10 AHA2 molecules per $\mu m^2$ PM area) (*Figure 3C*). A relatively sharp alteration of the AHA2-GFP amount was detected for the TZ (*Figure 3C*). If the amount of AHA2-GFP and BIR3-GFP molecules was set in ratio to the number BRI1-GFP molecules in the PM along the root tip axis, there was no alteration with respect to BIR3 (ratio: about 1.35), but a strong increase regarding AHA2 from 0.28 in the MZ to up to 5 in the late EZ.

Our significantly improved spatio-temporal refinement of previous data (*Pacifici et al., 2018*) by scRNA-Seq and quantitative CLSM demonstrate a coincidence of AHA2 protein accumulation with the onset of growth in the EZ. These results suggest that there may be a regulatory link between AHA2 protein accumulation and probably activity pattern and normal and BR-regulated root growth along the root tip axis. This hypothesis is particularly plausible given that AHA2 interacts physically with BRI1 and BAK1 and is phosphorylated within 5 min upon BR treatment in vivo (*Caesar et al., 2011*; *Ladwig et al., 2015*; *Lin et al., 2015*; *Yuan et al., 2018*).

## Modeling predicts the H⁺-ATPases being crucial regulators of the extracellullar pH in the BR/BRI1 response

To test the hypothesis formulated above, we decided to investigate the functional role of AHA in the context of BR-regulated signaling activity both experimentally and computationally. Here, we first sought to quantify and analyze the response in the EZ. With the key components of the $H^+$ homeostasis and nano-organized BRI1 complex quantified (see *Figure 3C*), we were able to tailor the model to represent a single epidermis cell in the EZ. By further using a combination of dose-response data and time-course measurements to fit the remaining unknown model parameters, we then should be able to analyze both the overall response and the temporal dynamics of the BR signaling module.

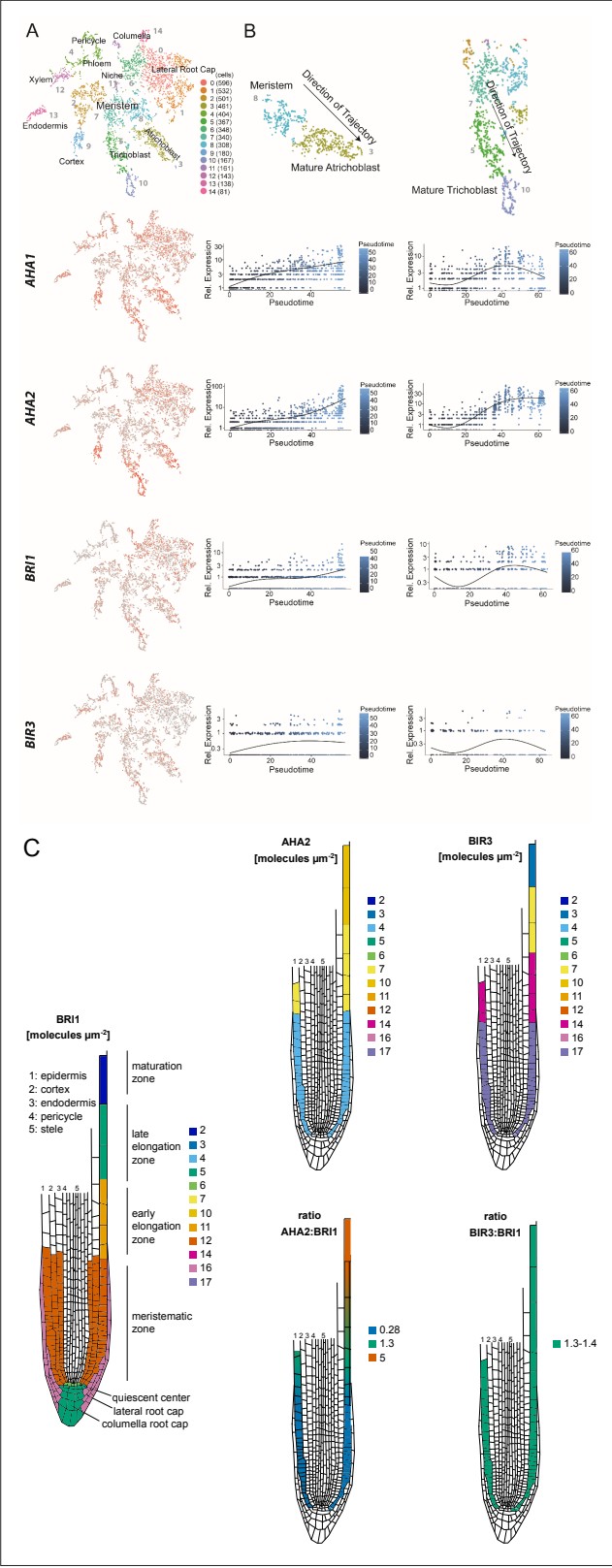

**Figure 3.** The constituents of the nano-scale organized BRI1 complex are spatio-temporally differentially expressed in the epidermal cells along the *Arabidopsis* root tip axis. (**A**) *AHA1*, *AHA2*, *BIR3* and *BRI1* transcript levels in the different cell types of the *Arabidopsis* root tip derived from scRNA-Seq data (***Ma et al., 2020***). The atrichoblasts and trichoblasts together represent the epidermal cells. (**B**) Developmental trajectories of *AHA1*,

*Figure 3 continued on next page*

*Figure 3 continued*

*AHA2, BIR3,* and *BRI1* transcript accumulation along the root tip (**Ma et al., 2020**). The transition from the MZ to the EZ is at a pseudotime value of around 30. (**C**) Upper panel. Number of the indicated GFP fusion proteins (molecules per µm²) in the plasma membrane of epidermal cells along the root tip axis. The values for BRI1-GFP and BAK1-GFP were taken from the literature (**Wilma van Esse et al., 2011**). Lower panel. The same but here the ratios of BRI1-GFP/AHA2-GFP and BRI1-GFP/BIR3-GFP molecules in the plasma membrane are given.

To measure the dose-response behavior and the time-course response to BR stimulation experimentally, we relied on the salt 8-hydroxypyrene-1,3,6-trisulfonic acid trisodium (HPTS), a non-invasive dye that incorporates into the plant cell wall and enables the ratiometric fluorescence readout of the pH conditions at cellular resolution (**Barbez et al., 2017**; **Appendix 1—figure 3**). To determine the apoplastic pH conditions 60 min after brassinolide (BL) application in the EZ, we performed a dose-response analysis. A significant decrease of the apoplastic pH was observed already at a BL concentration of 0.1 nM that continued up to a concentration of 10 nM (**Figure 4A**). Higher concentrations of BL did not further increase the cellular response in the EZ. This behavior is reproduced by the model ensemble (**Figure 4A**).

To capture not only the overall response to BL stimulation in the EZ but also its temporal dynamics, we further performed time-course measurements of the apoplastic pH in response to 10 nM BL using HPTS. Here, we observed a rapid acidification within 10 min after hormone application that is maintained for the remainder of the experiment (**Figure 4B**). This observation was again reproduced by the model ensemble (**Figure 4B**). At the same time, we could also capture the cell wall swelling in the model that has been observed in response to BL application previously (**Elgass et al., 2009**; **Caesar et al., 2011**; **Figure 4C**).

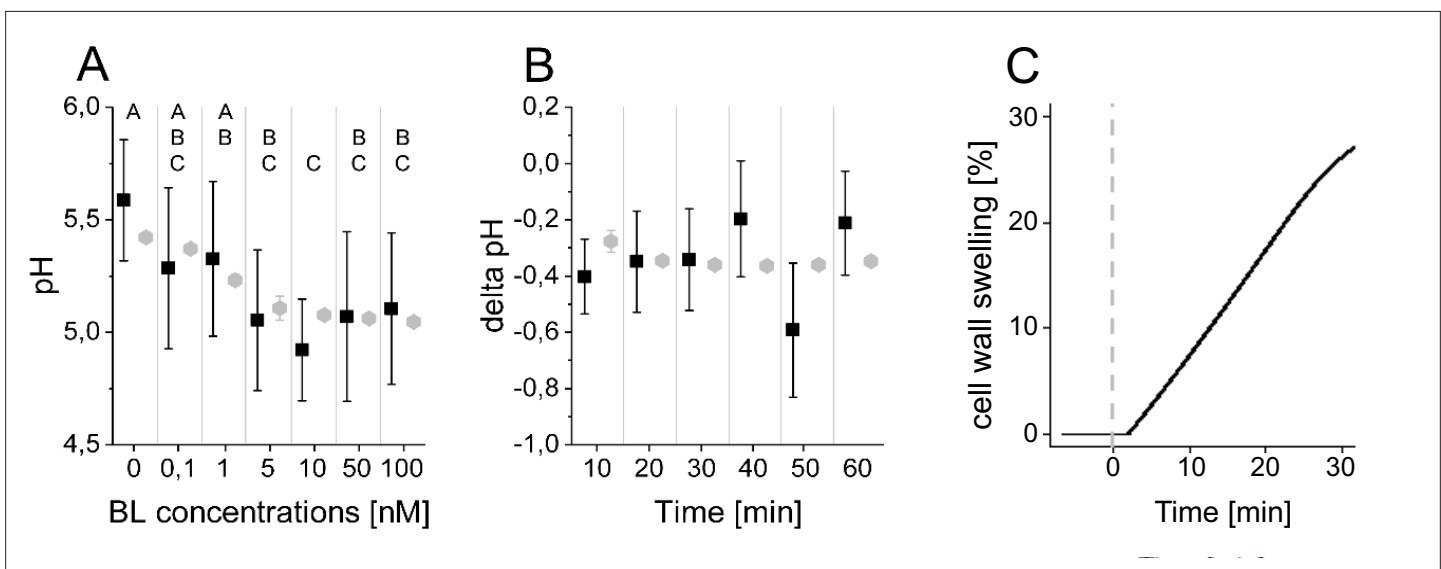

**Figure 4.** The computational model quantitatively and dynamically captures the sensitivity and kinetics of apoplastic acidification in *Arabidopsis* epidermal cells of the root EZ in response to BL. (**A**) HPTS-staining visualized (black quadrats) and computationally simulated (grey diamonds) dose-response behavior of apoplastic pH. Real or virtual BL incubation was done for 60 min. Error bars represent SD for the experimental data (n≥11) and the simulations of different model parameterizations (n = 10). Statistical evaluations to compare the effect of BL concentrations on experimental data, were performed by an One-way ANOVA followed by a Tukey-HSD post hoc test. Levels not connected by same letter are significantly different. The exact p-values can be found in the corresponding RAW data file. (**B**) HPTS-staining visualized (black quadrats) and computationally simulated (grey diamonds) time-course of apoplastic pH change in response to 10 nM BL. Error bars represent a corrected SD for the experimental data (n≥16) (for calculations see the corresponding RAW data file) and SD for the simulations of different model parameterizations (n = 10). Statistical evaluations on experimental data were performed as described in A. (**C**) Computationally simulated time course of relative wall swelling in response to 10 nM BL. The addition of BL at time 0 is indicated by the vertical dashed line.

The online version of this article includes the following source data for figure 4:

**Source data 1.** Raw data underlying the representation of the experimental results of *Figure 4*.

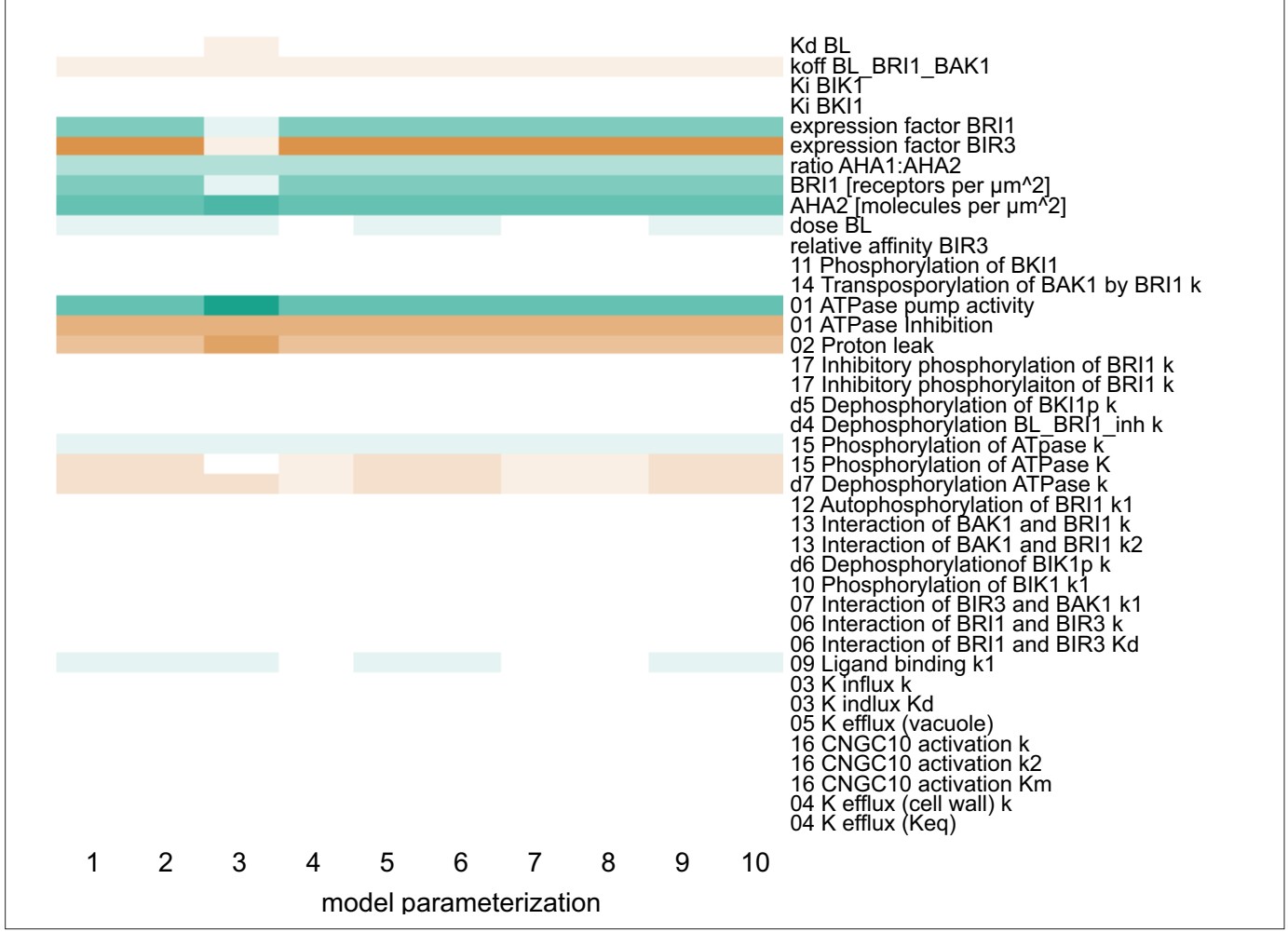

**Figure 5.** Computational calculation of scaled sensitivities of the cell wall acidification predicts AHA2 activity and molecules in the PM as well as BRI1 expression and molecules in the PM to be the deciding factors for the competence of *Arabidopsis* epidermal root cells to elongate in response to 5 min BL application for all parameterizations of the model. A positive influence is shown in green, a neutral in white and a negative in red, with the color saturation indicating the strength of the influence.

The online version of this article includes the following source data for figure 5:

**Source data 1.** Raw data underlying the representation of the results.

Using this model ensemble that specifically describes the behavior of a single epidermis cell in the EZ, we analyzed the importance of the individual model components and parameters for the cell physiological response by calculating the scaled sensitivities. In particular, this means that we calculated the relative change of the cell wall acidification in response to relative changes in model parameters while simulating the BR response stimulated with 10 nM BL for 5 min and 60 min. The results of the sensitivity analysis for all model parameterizations (n=10) are summarized in *Figure 5*, where a positive influence on the BR response is denoted in green, no influence is denoted in white and a negative influence is denoted by red, with the color saturation indicating the strength of the control. Notably, at the beginning of the BR response the initial concentrations of the receptor BRI1 and the proton pumps had a large impact. In addition, parameters influencing proton extrusion such as the degree of inhibition and the pump activity of the ATPases strongly controlled the early BR response across all model parameterizations (*Figure 5*). The sensitivities of the acidification 60 min after BL application in turn showed a greater control of down-regulating elements such as the inhibitory phosphorylation of the receptor (*Appendix 1—figure 4*), although the amount of proton pumps as well as their activity remained impactful. As our previous protein quantification data showed a near constant level of the receptor while the AHA2 levels change notably, this strongly supports the hypothesis that the proton

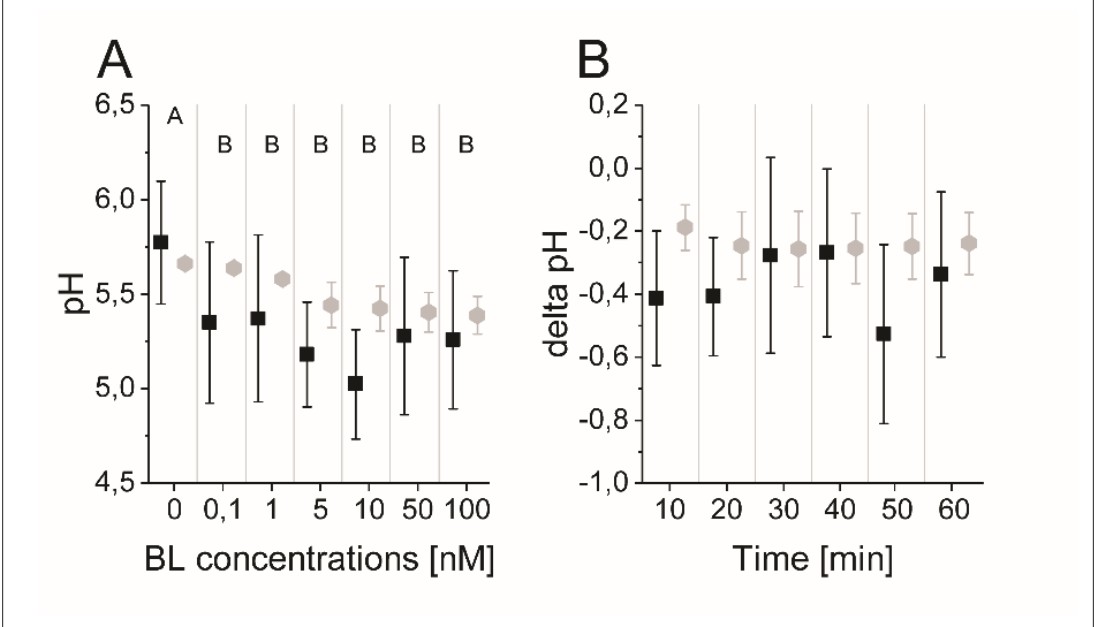

**Figure 6.** The model quantitatively and dynamically captures the sensitivity and kinetics of apoplastic acidification in *Arabidopsis* epidermal cells of the root MZ in response to BL. (**A**) HPTS-staining visualized (black quadrats) and computationally simulated (grey diamonds) dose-response behavior of apoplastic pH. Real or virtual BL incubation was done for 60 min. Error bars represent SD for the experimental data (n≥11) and the simulations of different model parameterizations (n = 10). Statistical evaluations on experimental data were performed as described in *Figure 4*. Levels not connected by same letter are significantly different. The exact p-values can be found in the corresponding RAW data file (**B**) HPTS-staining visualized (black quadrats) and computationally simulated (grey diamonds) time-course of apoplastic pH change in response to 10 nM BL. Error bars represent a corrected SD for the experimental data (n≥16) (for calculation see the corresponding RAW data file) and SD for the simulations of different model parameterizations (n = 10). Statistical evaluations on experimental data were performed as described in *Figure 4*.

The online version of this article includes the following source data for figure 6:

**Source data 1.** Raw data underlying the representation of the results.

pumps are the key elements that determine the competence of cells to respond to BR stimulation and react with elongation growth.

In consequence, the cells in the MZ should show a higher starting pH and react less strongly to BR stimulation due to the lower expression levels of AHA2. To predict the behavior of an epidermis cell in the MZ, we adjusted the model ensemble to instead represent a single epidermis cell in the MZ in terms of protein concentrations and compartment sizes. This model ensemble shows a higher resting pH and a reduced response to BR stimulation as evident in the dose-response behavior and kinetics properties that was supported in principle experimentally by HPTS visualization (*Figure 6*, *Appendix 1—figure 5*). However, due to the limitation in the sensitivity of the HPTS method and the biological variability in the different root preparations, the difference in the BL-induced acidification responses between MZ and EZ epidermal cells could not be captured statistically. The modeling approach is therefore advantageous for the prediction of small, cell physiological response differences which are difficult to establish experimentally due to high biological and methodological variability (*Appendix 1—figure 5*). Although the model captures the cellular physiology very well, we cannot entirely exclude the possibility that there is no difference between PH responses in the MZ and EZ cells.

## Experimental evaluation confirms the predicted relevance of the H⁺-ATPases for the extracellular pH control in the BR/BRI1 response

To confirm the predictions of the model experimentally, we used both HPTS and microelectrode ion flux estimation (MIFE) measurements. MIFE is another non-invasive experimental method in addition to HPTS measurements that allows for contact-free, real-time, electrophysiological measurements of H⁺ fluxes at the surface of roots by using an H⁺-specific electrode that mainly reflects the ATPase

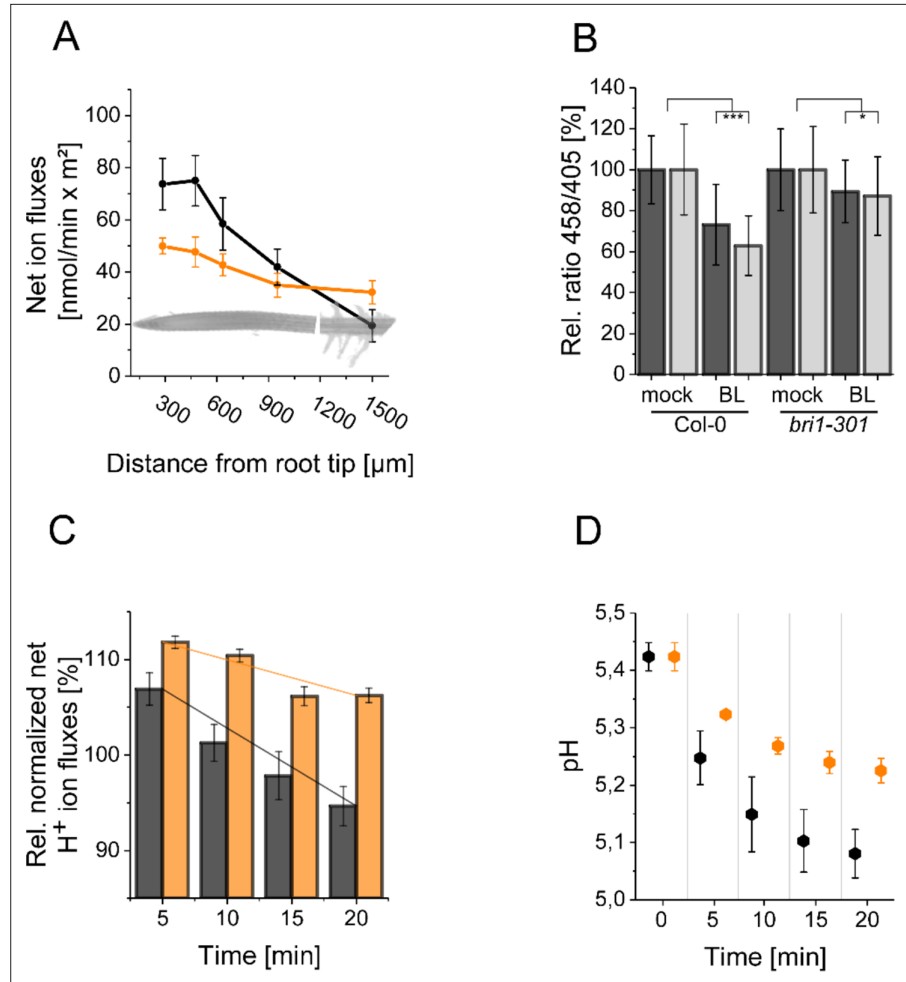

**Figure 7.** The resting apoplastic pH gradient of epidermal root cells along the axis and its regulation by BR depends on kinase-active BRI1. (**A**) MIFE recording of the H⁺ fluxes along the root axis of *Arabidopsis* wild type (black line) and *bri1-301* mutant (orange line) plants. Measurements were performed from 250 μm of the root tip off to the root hair zone. Error bars represent SD (n = 3). (**B**) Comparison of the relative apoplastic pH (ratio 458/405) of epidermal root cells in the MZ (black bars) and EZ (grey bars) of wild type and *bri1-301* mutant plants after 60 min of BL (10 nM) or mock treatment, visualized by HPTS staining. The data derived from the mock treatments of the respective line were set to 100. Error bars represent SD (n≥30). Statistical evaluations were performed by comparing the respective groups separately (e.g. 'Col-0 MZ mock' compared with 'Col-0 MZ BL'). Depending on the distribution of data and other assumptions either a (pooled) Two-Tailed T-Test or a Two-Tailed Wilcoxon Test were applied. The black asterisks indicate statistically significant differences (***: p≤0.001); (*: p≤0.05). The exact p-values can be found in the corresponding RAW data file. (**C**) Relative H⁺ fluxes at the EZ of wild type (black bars) and *bri1-301* mutant (orange bars) plants between 5 and 20 min after application of 10 nM BL recorded by MIFE. The flux directly before the addition of BL was set to 100. The increase in net influx after treatment is due to a disturbance of the H⁺ conditions at the root surface, which is observed with any treatment. The solid lines illustrate the linear regression. The slope is –0.818 for the wild type and –0.371 for *bri1-301*. Error bars represent SD (n = 3). (**D**) Simulated response to 10 nM BL for the wild type (black) and *bri1-301* mutant (orange), under the assumption that the *bri1-301* mutant is biochemically half as active as the wild type. Error bars represent SD (n=10).

The online version of this article includes the following source data for figure 7:

**Source data 1.** Raw data underlying the representation of the results.

activity in the underlying tissues (**Newman, 2001**; **Fuglsang et al., 2014**). Confirming previous results (**Staal et al., 2011**), our MIFE measurements along the *Arabidopsis* root tip revealed a net H⁺ influx at the MZ, which then was drastically reduced in the EZ implying higher H⁺ ATPase activity in this region (**Figure 7A**). These differential H⁺ fluxes translate into a pH gradient along the surface of the root tip with the MZ less acidic and the EZ more acidic (**Staal et al., 2011**). Using HPTS, we substantiated

the MIFE results and confirm the observation of *Barbez et al., 2017* that there is an apoplastic pH gradient of the epidermal root cells from the MZ (less acidic) to the EZ (more acidic) (*Appendix 1— figure 3*).

To address the question whether the establishment of the resting pH gradient and the differential changes of the pH conditions upon external BL application depend on fully functional BRI1, we used the *bri1-301* mutant for further HPTS and MIFE measurements. In the *bri1-301* mutant a BRI1 version with a reduced kinase activity is expressed, which causes a weak defective root growth phenotype at ambient temperature (*Lv et al., 2018*; *Zhang et al., 2018*). This less-pronounced *bri1-301* phenotype allows HPTS and MIFE measurements technically comparable to those of wild type plants. As shown in *Figure 7B*, the BL-induced changes in the apoplastic pH - here represented in the relative change of 458/405 fluorescence emission ratio - observed for wild type were significantly reduced in the *bri1-301* mutant. The HPTS data were again supported by our MIFE measurements: The wild type cells of the EZ showed an increase in the net $H^+$ efflux upon application of 10 nM BL, which continued over the measurement period of 20 min, whilst the cells of the *bri1-301* mutant responded much less (*Figure 7C*). Under the assumption that the mutant BRI1-301 receptor is biochemically half as active as wild type BRI1 the model is able to capture the experimentally measured behavior correctly (*Figure 7D*).

In summary, the concordant results of our experimental approaches including those of *Caesar et al., 2011* substantiate the prediction of the mathematical model that the enhanced level of $H^+$-ATPase amount and activity in relation to the number of BRI1 receptors define the BR-regulated apoplastic acidification and linked hyperpolarization of the $E_m$. Moreover, the maintenance of the pH gradient and $H^+$-fluxes along the root tip axis and the BL regulation of alterations depend on kinase-active BRI1.

## Modeling predicts a cation channel for charge compensation during $H^+$ export and PM hyperpolarization

The great value of mathematical modeling and prediction is especially demonstrated after we calculated the membrane potential derived from the pH value changes in the apoplastic cell space of the root tip upon BL treatment and compared it with the previously experimentally determined $E_m$ changes (*Caesar et al., 2011*). The calculated $E_m$ change induced by the change in charge distribution due the acidification of the apoplastic space was much stronger than the measured one (*Figure 8A* and Appendix 1 - example calculation of $E_m$ and pH change based on membrane area, specific membrane capacitance and transported charges): An acidification from pH 5.4 to pH 5.0 in response to 10 nM BL corresponds to an $E_m$ change of approximately 28 mV, as opposed to the experimentally measured 7.2 mV (*Caesar et al., 2011*). As mentioned before (see *Figure 2*) and according to the prediction of our model, this discrepancy values was eliminated, if an import of monovalent cations such as potassium ($K^+$), which predominantly contributes to the $E_m$ of the PM in plant cells (*Higinbotham, 1973*), took place in parallel to the ATPase generated $H^+$ extrusion. Against the background that BAK1 and AHA2 interact with a cation channel of the cyclic nucleotide-gated ion channel (CNGC) family in the phytosulfokine receptor 1-mediated growth response (CNGC17; *Ladwig et al., 2015*), we searched in the literature and the *Arabidopsis* eFP browser (*Sullivan et al., 2019*) for a CNGC member, which is expressed in the root tip, localizes to the PM and imports monovalent ions, and is functionally linked to cell expansion. Applying these criteria, we identified CNGC10 as a potential candidate. Although to a low extent, CNGC10 is expressed in all cell types of the root tip (*Brady et al., 2007*; *Jin et al., 2015*; *Ma et al., 2020*), localizes to the plasma membrane, transports $K^+$ and is functionally linked to cell expansion (*Borsics et al., 2007*; *Christopher et al., 2007*; *Duszyn et al., 2019*). When CNGC10 and its $K^+$ transport properties derived from the literature above were integrated into our model, the discrepancy between the calculated and measured value was gone (*Figure 8B*). This suggests that the CNGC10-mediated influx of potassium can principally counteract the ATPase-caused efflux of $H^+$ into the apoplast in the root tip.

To test whether CNGC10 is able to interact with components of the nano-organized BRI1 complexes such as BRI1, BAK1 and AHA2, Förster resonance energy transfer by fluorescence lifetime imaging (FRET-FLIM) analyses in transiently transformed *Nicotiana benthamiana* leaf cells and yeast mating-based split-ubiquitin (mbSUS) assays were performed. The growth of yeast cells on interaction selective media and the reduction of the GFP fluorescence lifetime (FLT) revealed a spatially very close

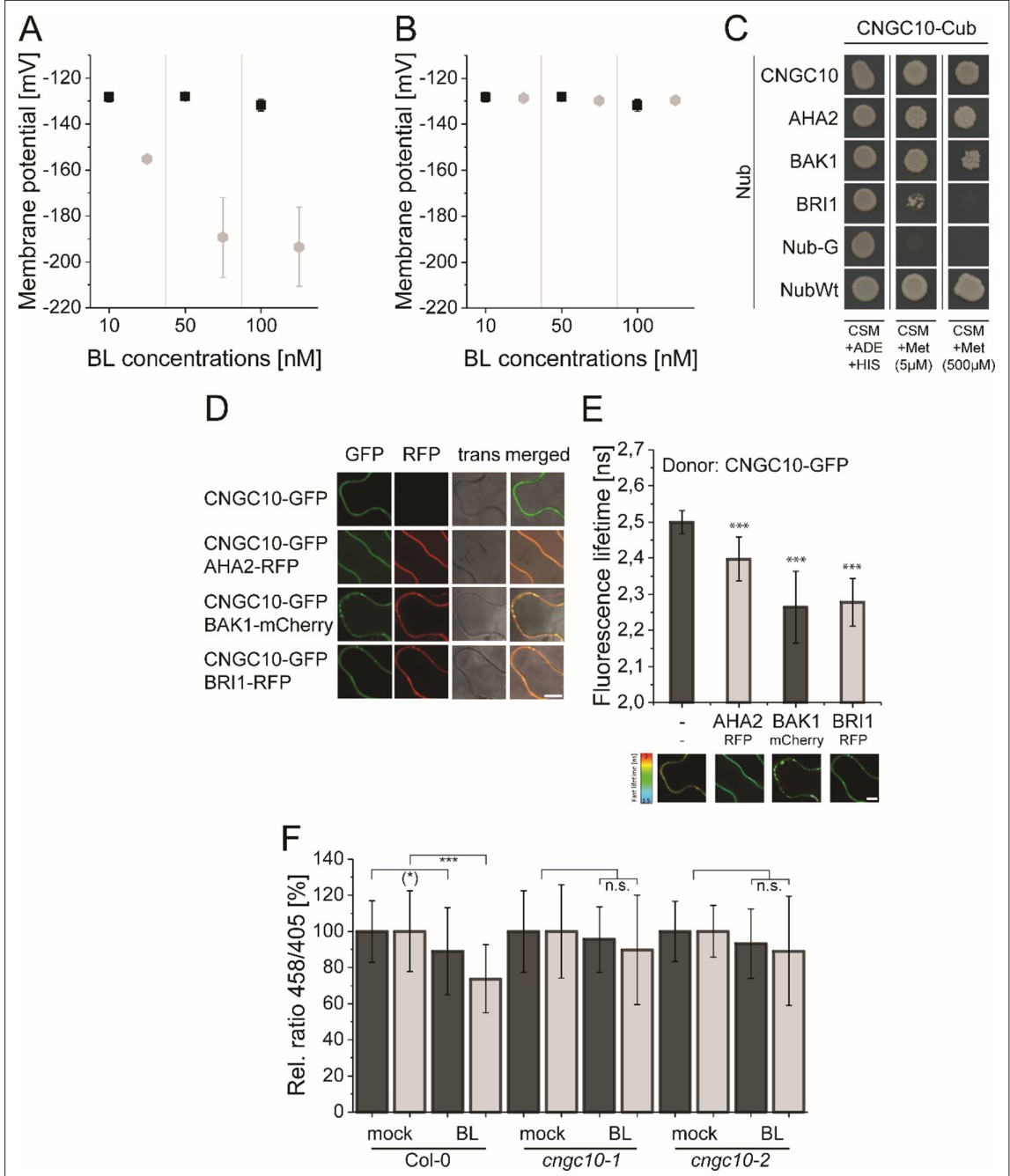

**Figure 8.** The computational model predicts the existence of a potassium channel, likely to be CNGC10, to maintain the homeostasis of the plasma membrane potential and apoplastic pH in *Arabidopsis* epidermal root cells. (**A**) Modeled $E_m$ in the presence of different BL concentrations without the integration of potassium import (grey diamonds) in comparison to the published experimental data [black quadrats; *Caesar et al., 2011*] after 20 min of BL treatment. (**B**) Modeled $E_m$ in the presence of different BL concentrations with the integration of the CNGC10 potassium channel (grey diamonds) in comparison to the published experimental data [black quadrats; *Caesar et al., 2011*]. Error bars in A and B represent SEM (n≥4) in the experimental approach and SD (n = 5) of simulation results of the different model parameterizations. (**C**) CNGC10 forms homomers and interacts with BAK1 and AHA2 in the yeast mating-based split-ubiquitin system. The indicated combinations of Cub and Nub fusion constructs were transformed into yeast cells. Yeast cells were then grown either on media selective for the presence of the plasmids (CSM +Ade, +His) or on interaction selective media with two different concentrations (5 μM, 500 μM) of methionine (CSM +Met). The combination of CNGC10-Cub with Nub-G served as negative and that with NubWT as positive control. (**D**) CNGC10 colocalizes with AHA2, BAK1, and BRI1 in the plasma membrane of plant cells. Representative confocal images of transiently transformed tobacco epidermal leaf cells expressing the indicated fusion proteins. The Scale bars represents 10 μm. (**E**) CNGC10 is spatially closely associated with AHA2, BAK1 and BRI1 in the plasma membrane of plant cells. Fluorescence lifetime imaging microscopy (FLIM) analysis comparing the different Förster resonance energy transfer (FRET) pairs. Top: FLIM measurements of transiently transformed tobacco epidermal leaf cells

*Figure 8 continued on next page*

*Figure 8 continued*

expressing the CNGC10-GFP donor fusion with the indicated RFP or mCherry acceptor fusions. Error bars indicate SD (n≥20). Statistical evaluations were performed by a Kruskal-Wallis Test followed by Steel-Dwass post hoc test. The black asterisks indicate statistically significant differences (***: $P \leq 0.0001$). Bottom: Heat maps of representative plasma membrane areas used for FLIM measurements. The donor lifetimes of CNGC10 are color-coded according the scale at the left. The scale bar represents 7 µm. (**F**) Comparison of the relative apoplastic pH (ratio 458/405) of epidermal root cells in the MZ (black bars) and EZ (grey bars) of wild type and two independent *cngc10* mutant lines after 60 min of BL (10 nM) or mock treatment, visualized by HPTS staining. The data derived from the mock treatments of the respective line were set to 100. Error bars represent SD (n≥27). Statistical evaluations were performed as described in *Figure 7B*. The black asterisks indicate statistically significant differences (***: $p \leq 0.0001$); ((*): $p = 0.0603$; borderline p-value); n.s.: not significant. The exact p-values can be found in the corresponding RAW data file.

The online version of this article includes the following source data for figure 8:

**Source data 1.** Raw data underlying the representation of the results.

association (below 13 nm; *Glöckner et al., 2020*) and interaction, respectively, of CNGC10 with BRI1, BAK1, and AHA2 (*Figure 8C–E*). To test whether CNGC10 functions in the fast BR response pathway, we analyzed the BL-induced apoplastic pH change in two independent *cngc10* loss-of-function lines (*Jin et al., 2015*; *Borsics et al., 2007*) compared to the corresponding wild type (Col-0). In contrast to the wild type both mutants did not acidify the apoplast of the cells in the MZ and EZ upon application of 10 nM BL (*Figure 8F*). These data indicate that CNGC10 is the major K+ channel to maintain the $E_m$ homeostasis of the PM during BL-induced apoplastic acidification primarily in the EZ and appears to be an additional constituent of the elongation growth-related, nano-scale organized BRI1 complexes.

## Discussion

BRs fulfill a central role in regulating plant physiology, growth and development as well as adaption to the environment (*Lv and Li, 2020*). A prominent example for a BR function is the rapid initiation of the (epidermal) cell growth in the EZ but not in the MZ of the *Arabidopsis* root tip (*Lv and Li, 2020*). Evidently, the hormone acts on an already existing functional competence of the root cells that, according to our experimental data, cannot be attributed to the absence of the BRI1/BAK1 perception system but must have other reasons. *Pavelescu et al., 2018* proposed that BRI1 signaling in the MZ is sufficient for root growth. More recent, complementary data show the highest BR concentration in the EZ, where it overlaps with BR/BRI1 signaling maxima with respect to cell elongation (*Vukašinović et al., 2021*). This observation implicates BRI1-dependent BR perception and signaling in the regulation of cell elongation in the EZ as well. Moreover, although the main molecular determinants of BR perception and signaling are known, the processes leading to this competence and its realization towards, in this case, elongation were so far not well understood.

By an iterative combination of computational modeling and wet lab experiments, we addressed this question by analyzing the dynamics of the PM-resident fast BR response pathway as a whole. The model's predictions of the crucial constituents in the nano-organized BRI1 complexes were experimentally verified, thereby determining the deciding and regulating elements for the signaling output. Using a detailed kinetic model on the basis of ODEs, we could analyze the interplay of the signaling components and the system as a whole: We captured the dynamics of the apoplastic acidification and $E_m$ hyperpolarization without BR and in response to the hormone. In addition, we showed that the rapidity and degree of the apoplast acidification in response to BR application is determined largely by the amount and activity of the ATPase AHA2 in the PM of the epidermal root cells. Furthermore, the model predicted that an influx of cations is required in order to explain both the pH and $E_m$ changes of the PM simultaneously. We found that CNGC10 is the responsible cation (potassium) channel. Besides functional evidence, it associates with BRI1, BAK1 and AHA2 in vivo. CNGC10 could therefore be another constituent of nano-organized BRI1 complexes in the PM of root cells.

If we project the measured AHA2 amount and AHA activity, and the apoplastic pH of epidermal cells along the axis of the root tip, we observe that they both increase and decrease, respectively, with the beginning of the EZ and strongly correlate with the competence to grow upon BL application. Proposed by the computational model, AHA2 appear to be the rate-limiting factor for the cells' competence to respond to BR by short-term cell physiological responses and eventually elongation. This gradient of AHA2 expression implies that the BR implements on an already existing, functional competence of the cells along the root axis that cannot be attributed to the absence of

the BRI1/BAK1 perception system. The competence to respond to the hormone is rather reflected by a gradient of AHA2 accumulation and probably differential AHA2 incorporation into nano-organized BRI1 complexes along the root axis. As reported previously, the establishment of the AHA2 accumulation along the root axis is achieved by the interplay of cytokinin and auxin activity during root development (*Pacifici et al., 2018*). This agrees with the suggestion of our model that the cells of the MZ should exhibit elongation growth if AHA2 is ectopically expressed and thus acidification enhanced. This is actually the case: Inducible expression of AHA2 enhances the length of MZ cells but in parallel reduces MZ cell number during root development (*Pacifici et al., 2018*). Another competence pattern was recently reported to be the differential local BR biosynthesis along the root axis (*Vukašinović et al., 2021*): While BRs are present throughout the root, the expression of BR synthesis enzymes is highest in the EZ. Remarkably, the additional application of exogenous BR in high concentration causes the elongation of MZ cells and decrease in MZ cell number (*Vukašinović et al., 2021*), copying the phenotype caused by the inducible expression of AHA2 in the MZ zone (*Pacifici et al., 2018*). This suggests that two different competence pattern, BR biosynthesis and accumulation of AHAs, superimpose along the root axis. Whether the differential expression of BR synthetic genes along the root axis is also controlled by the interplay of cytokinin and auxin, has to be analyzed in the future.

The final output of the cell elongation appears to require the sequence of short-term (within minutes) and long-term response mechanisms (from several hours to days). According to our modelling and experimental data as well as published results, short-term molecular and cell physiological responses to BR are predominantly linked to the rapid activation of AHAs very likely by their phosphorylation at two residues in the large cytoplasmic loop (Ser315 and Thr328 in AHA2) within 5 min (*Lin et al., 2015*), followed by the acidification of the apoplastic space detectable within 10 min (shown here), hyperpolarization of the PM detectable within 10 min (*Elgass et al., 2009*; *Caesar et al., 2011*) and wall swelling detectable within 20 min (*Elgass et al., 2009*; *Caesar et al., 2011*). Based on the acid growth theory, these AHA activity-related responses are the prerequisite for the onset of cell elongation (*Cosgrove, 2000*; *Hager, 2003*). Therefore, we propose that the ongoing of the BR-regulated elongation growth, that involves altered gene expression later in time, is not possible if the initial rapid processes do not occur adequately. The BR-mediated control of the $H^+$ ATPase and, thus, the $E_m$ concerns not only elongation growth.

The $E_m$ is also central for adaptive responses to a broad range of abiotic cues and for developmental processes. Our observations therefore suggest that the regulation of $H^+$ ATPase might contribute to the versatile functions of BR in many of these processes (*Lv and Li, 2020*; *Wolf, 2020*).

The BR-induced cell physiological processes occurring in the minute range appear to require higher hormone concentrations (around 1–10 nM) compared to those for long-term root growth and (other) gene expression-related processes in the range of hours or days (0.1–0.25 nM; *Chaiwanon and Wang, 2015*; *Vukašinović et al., 2021*). This discrepancy can have different reasons: Firstly, the physiologically effective concentration in the root tissue after short-time BL treatment is not known to us. Due to the short time for diffusion into the root, it may well be that the BL concentration at the target tissues is lower than the externally applied one. Secondly, it cannot be excluded that the continuous growth of seedlings on media containing very low BR concentrations induces the accumulation of BR itself or other growth-promoting hormones in the long-term, for instance by the enhanced expression of their biosynthetic genes. Interestingly, the short-term sequence of events in the *Arabidopsis* root tip is significantly faster and requires lower BL amounts than the AHA activation during cell elongation in the *Arabidopsis* hypocotyl. There, phosphorylation of AHAs at the penultimate threonine (Thr947 in AHA2) is detectable 60 min after BL application at the earliest and at BL concentration of at least 100 nM (*Minami et al., 2019*). Furthermore, the phosphorylation of the penultimate threonine in AHAs appears not to be required for at least the BL-induced $E_m$ hyperpolarization in tobacco leaf cells 30 min after application of 10 nM BL (*Witthöft et al., 2011*). Moreover, the enhanced phosphorylation of the AHAs' penultimate Thr by BR - measured 2 hr after application of 1 µM BL - involves the interaction of SAUR15 with BRI1 (*Li et al., 2022*). It is therefore tempting to speculate that a cascade of different phosphorylation events might be involved in the temporal regulation of AHA activity in different plant organs.

As discussed above, we propose an increased number of AHA2-containing nano-organized BRI1 complexes or an enhanced proportion of AHA2 therein from the MZ to the EZ cells. Varying the composition of nano-organized receptor complexes along a developmental gradient is an elegant way

to achieve cell- and tissue-specific responses to a given cue, when the number of available perception, signaling and output elements is limited. This principle seems to be realized in various BRI1-mediated functions. For example, the BRI1-dependent regulation of the vascular cell fate in the MZ of the root or the BRI1-mediated cross-tissue control of the cell wall homeostasis require nano-organized BRI1 complexes that contain at least additionally RLP44 (*Wolf et al., 2014*; *Holzwart et al., 2018*). More-over, RLP44-containing BRI1/BAK1 nanoclusters are spatially distinct from for instance FLS2/BAK1 nanoclusters (*Glöckner et al., 2020*).

The availability of a sophisticated model also enables in silico genetics that simplify the understanding of complex regulatory processes and their sometimes non-intuitive effects on the functional outputs. This is illustrated by the example of the negative regulator BIR3 that prevents the interaction of BAK1 and BRI1 in the absence of the hormone thereby suppressing BR signaling (*Imkampe et al., 2017*; *Großeholz et al., 2020*). Our computational model not only represents the previously published BR activity of the growth-related phenotypes of the *Arabidopsis bir3* mutant and BIR3-overexpressing plants but also allows predictions about the dose-dependent fine-tuning of BIR3 on BR/BRI1/BAK1-related functions (see *Appendix 1—figure 6*). Such in silico genetic and physiological approaches can be used to determine the functional and regulatory significance of other components of the fast BR response pathway as shown for AHA2 and the prediction of a cation channel for charge compensation. Thus, computational modeling facilitates the prioritization of the components of a perception and signaling system whose function should first be tested experimentally.

In summary, the recurrent application of computational modeling and subsequent wet lab experiments provided a novel in-depth and quantitative view of the initial cell physiological processes, regulatory networks and information processing leading to a minimal molecular and biochemical framework linked to BR-regulated elongation growth along the axis of the root tip. This approach can in principle be applied for the analysis of every signal perception and transduction process as long as a minimal set of elements and quantitative data are available or experimentally accessible, as has been demonstrated for example in the in-depth analysis of the PLT-auxin network during root zonation development in *Arabidopsis thaliana* (*Salvi et al., 2020*; *Rutten and Ten Tusscher, 2021*).

The ongoing challenge will now be to establish a model of anisotropic elongation growth along all tissues of the root tip, as it was initiated for the description of BR-regulated radial growth of the root MS (*Fridman et al., 2021*). At the cellular level, the further aim is to expand and refine the model by the integration of the data of the potentially BR-modified composition, assembly and dynamics of the nano-organized BRI1 complexes in the PM obtained by sophisticated super-resolution microscopy and in vivo FRET studies (*Glöckner et al., 2020*).

# Methods and materials
## Experimental methods
### Plant material
Seeds of the *Arabidopsis* mutants and lines expressing the different fusion proteins were surface sterilized and placed on ½ Murashige and Skoog (MS) medium plates with 1% phytoagar and 1% sucrose followed by stratification at 4 ° C in the dark for 2 days. Afterwards the plants were grown in growth chambers at 20 ° C under long day conditions (16 hr light/8 hr dark) for 5 days. The transgenic *Arabidopsis* lines (Col-0 ecotype) contained either a *pBRI1:BRI1-GFP* (wild type background; *Friedrichsen et al., 2000*), a *pAHA2:AHA2-GFP* (aha2-4 mutant background; *Fuglsang et al., 2014*) or a *pBIR3:BIR3-GFP* construct (bir3-2 background; *Imkampe et al., 2017*). The *Arabidopsis bri1-301* mutant (Col-0) was described in detail previously (*Lv et al., 2018*; *Zhang et al., 2018* and references therein). The previously described *Arabidopsis cngc10-1* and *cngc10-2* mutants (Col-0) (*Jin et al., 2015*; *Borsics et al., 2007*) were obtained from the Nottingham stock center (SALK_015952, SALK_071112).

### Confocal imaging
Quantification of the GFP signal on five days old seedlings was performed by confocal laser scanning microscopy (CLSM) on a SP8 laser scanning microscope (Leica Microsystems GmbH) with HyD detectors and a HC PL APOCS2 63 x/1.20 WATER objective. Detection range was set to 500 nm – 540 nm with 400 V gain and line averaging of 4. An adequate laser power for the 488 nm laser was applied

to avoid the saturation of the signal and to ensure a dynamic range across the expression levels of the different transgenic plants. The identical excitation and detection settings were used for all image quantifications. In six imaging sessions, ten straight lying root tips were imaged in the following way: The root tip was placed to the left border of a 1024x512 pixel image. The images for the quantification were taken in a way that 4–5 lanes of epidermal cells were in focus. Fluorescence intensity was quantified with a 50x50 µm region of interest (ROI) in Fiji/Image J. The ROI had to be completely filled by the fluorescence signal, hence "too high" z-layer-images, not filling the ROI completely, were excluded. Also, not completely straight-lying roots were excluded, so that a total of 40 measurements per transgenic plant line were finally used. As readout the Integrated Intensity Feature of Fiji, summing up the intensity of all pixels in a ROI, was used. For statistics, all measurements of 40 roots of at least three plant lines were combined.

## Microelectrode ion flux estimation (MIFE) measurement

For MIFE measurements, 5-day-old seedlings were grown as described but in continuous light. Experiments were performed as described by *Fuglsang et al., 2014*. The seedlings were equilibrated in bath medium (0.1 mM $CaCl_2$, 0.5 mM KCl, pH 5.8) for 2 h before the measurements. Only seedlings without proton oscillations were used. At time point 0.1 nM BL was added. The bathing solution was mixed two times by carefully pipetting up and down after addition of BL. The proximal position of the electrode (near the root) and the distal position (far from the root) were swapped compared to the previous study (*Fuglsang et al., 2014*). Consequently, a decrease in values represents proton efflux and an increase represents proton influx in our measurements.

## 8-Hydroxypyrene-1,3,6-trisulfonic acid trisodium salt (HPTS) measurement

For root apoplastic pH measurements, plates containing ½ MS agar media pH 5.7 without buffer, 1 mM HPTS dye, and the respective treatments were used. Five-day-old *Arabidopsis* seedlings were transferred onto the media and treated for 60 min with HPTS prior to imaging. For shorter treatments, seedlings were prestained with HPTS and subsequently treated according to the indications. For imaging, the plants on the media were flipped into a nunc imaging chamber (Ibidi 80286), the roots being close to the chamber bottom and covered by the media. Ratiometric imaging was conducted at an inverted Zeiss LSM880 confocal scanning microscope. The 405 nm and 458 nm laser were used at 0.2% and 100% intensity respectively, a PMT detector range from 495 to 535 nm was used and line sequential scans were performed. The detector gain was set at 1200. For imaging, a 40 x water immersion objective was used. The evaluation of ratio in the resulting images was determined following the workflow described by *Barbez et al., 2017*. For calibration curve measurements, ½MS agar media supplemented with 10 mM MES were adjusted to the desired pH and roots of 5-day-old seedlings were analyzed as described above.

## Mating-based split-ubiquitin system (mbSUS) measurements

For the mbSUS the coding sequences of *CNGC10*, *AHA2*, *BAK1* and *BRI1* were either fused to the sequences coding for the C-terminal part of ubiquitin (Cub) or the N-terminal part of ubiquitin (Nub). Namely, the plasmids pMetYC (Cub) and pXNubA22 (Nub) were used (*Grefen et al., 2009*). pNubWt-Xgate (*Obrdlik et al., 2004*) and the empty pXNubA22 vector served as positive and negative control, respectively. The experiments were performed as described by *Grefen, 2014* with some modifications: After dropping the mated yeasts on yeast extract peptone dextrose (YPD) plates they were scratched off with pipette tips, resuspended in 100 µl $H_2O$ and 5 µl were transferred to complete supplement mixture (CSM)-Leu -Trp -Ura -Met plates. The growth assay was performed with adjusted optical density of the yeast cultures in one dilution. Here, vector selective plates (CSM-Leu -Trp -Ura -Met) or interaction selective plates (CSM-Leu -Trp -Ura -Met, -Ade, -His) with 5 µM and 500 µM methionine were used. The growth of the yeast was documented after 72 hr of incubation at 28 °C.

## FRET-FLIM analysis

For FRET-FLIM analysis, the coding sequences were expressed as C-terminal fluorophore fusions, using pH7FWG2 (GFP), pB7RWG2 (RFP), or pABind-mCherry (*Karimi et al., 2002*; *Bleckmann et al., 2010*). These binary vectors and *p19* as gene silencing suppressor were transformed into *Agrobacterium tumefaciens* strain GV3101 and infiltrated into *Nicotiana benthamiana* leaves. The

measurements were performed 2–3 days after infiltration using an SP8 laser scanning microscope (Leica Microsystems GmbH) with LAS AF and SymPhoTime (PicoQuant) software as described (*Veerabagu et al., 2012*). Before performing the FRET-FLIM measurement, the presence of the fluorophores was imaged by using 488 nm or 561 nm lasers for GFP or RFP excitation, respectively. The fluorescence lifetime $\tau$ [ns] of either the donor only expressing cells or the cells expressing the indicated combinations was measured with a pulsed laser as an excitation light source with 470 nm and a repetition rate of 40 MHz (PicoQuant Sepia Multichannel Picosecond Diode Laser, PicoQuant Timeharp 260 TCSPC Module and Picosecond Event Timer). The acquisition was performed until 500 photons in the brightest pixel were reached. To obtain the GFP fluorescence lifetime, data processing was performed with SymPhoTime software and bi-exponential curve fitting and correction for the instrument response function.

## Statistics

All statistical evaluations were performed with SAS JMP 14. The applied tests are indicated within the respective figure texts. Detailed information about the statistics evaluations can be found in the RAW data files.

## Computational methods

### Model setup

The model consisting of ordinary differential equations was constructed in COPASI (*Hoops et al., 2006*; *Mendes et al., 2009*) 4.30, build 240, running on a 64-bit machine with Windows 8. Reactions were defined as mass action or Michaelis Menten kinetics where appropriate (see *Appendix 1—table 3*). Compartment sizes and parameters were defined based on experimental data if possible (*Appendix 1—Tables 1 and 3*). Unknown parameters were determined by parameter estimation. The schematic of the model was drawn using VANTED (*Junker et al., 2006*) and adheres to the Systems Biology Standard of Graphical Notation (SBGN) (*Novère et al., 2009*).

### Parametrization

All unknown model parameters, where no or only a range of experimental information were available, were estimated. To account for parameter non-identifiabilities, we generated 10 independent parameter sets by randomly sampling the starting parameter values before running the parameter estimation. Each parameter estimation run was set up using the particle swarm algorithm as implemented in COPASI 4.30 (*Hoops et al., 2006*), using 5,000 generations with a swarm size of 50 individual parameter combinations. The parameter estimation was repeated until the resulting solution had a $\chi^2$ around 10.45.

### Model analyses

The time-course simulations were run deterministically using the LSODA algorithm as implemented in COPASI. The simulations of the *bri1-301* mutant were run by setting all rate constants of phosphorylation reactions catalyzed by BRI1 to ½ the original value. The relevant reactions were $r_{10}$, $r_{11}$, $r_{12}$, $r_{14}$, $r_{15}$, and $r_{16}$. The impact of different BIR3 concentrations was analyzed using the parameter scan task in COPASI to simulate the time course of the pH over the time frame of 20 min. The scaled sensitivities of the extracellular pH change in response to changes in model parameters were calculated as $scaled\,sensitivity = \frac{ln(delta\,pH)}{ln(P_i)}$ at 5 min and 60 min. Results were plotted using R (*R CoreTeam, 2020*).

## Acknowledgements

The research in our laboratories is supported by the German Research Foundation (DFG) with grants to KH, KS and UK (CRC 1101-A02/D02) and grants for scientific equipment (INST 37/819–1 FUGG, INST 37/965–1 FUGG, INST 37/991–1 FUGG, INST 37/992–1 FUGG). We also thank the Schmeil Foundation and the Joachim Herz Stiftung for their support of RG. In addition, we are grateful for the support by Tom Denyer, ZMBP - University of Tübingen, for his help in the interpretation of the scRNA-Seq data.

## Additional information

### Funding

| Funder | Grant reference number | Author |
|---|---|---|
| Deutsche Forschungsgemeinschaft | CRC 1101 | Karin Schumacher |
| Deutsche Forschungsgemeinschaft | INST 37/819- 594 1 FUGG | Klaus Harter |
| Schmeil Stiftung | RG | Ruth Großeholz |
| Joachim Herz Stiftung | RG | Ruth Großeholz |
| Deutsche Forschungsgemeinschaft | INST 37/965-1 FUGG | Klaus Harter |
| Deutsche Forschungsgemeinschaft | INST 37/991-1 FUGG | Klaus Harter |
| Deutsche Forschungsgemeinschaft | INST 37/992-1 FUGG | Klaus Harter |

The funders had no role in study design, data collection and interpretation, or the decision to submit the work for publication.

### Author contributions

Ruth Großeholz, Conceptualization, Data curation, Software, Formal analysis, Funding acquisition, Validation, Investigation, Visualization, Methodology, Writing – original draft, Writing – review and editing; Friederike Wanke, Conceptualization, Data curation, Formal analysis, Supervision, Validation, Investigation, Visualization, Methodology, Writing – original draft, Writing – review and editing; Leander Rohr, Formal analysis, Validation, Investigation, Visualization, Methodology, Writing – review and editing; Nina Glöckner, Conceptualization, Data curation, Formal analysis, Validation, Investigation, Visualization, Methodology, Writing – original draft, Writing – review and editing; Luiselotte Rausch, Data curation, Formal analysis, Investigation, Visualization; Stefan Scholl, Data curation, Formal analysis, Validation, Visualization, Methodology; Emanuele Scacchi, Conceptualization, Supervision, Validation, Methodology, Writing – review and editing; Amelie-Jette Spazierer, Data curation, Validation, Investigation; Lana Shabala, Resources, Formal analysis, Supervision, Methodology; Sergey Shabala, Resources, Formal analysis, Supervision, Validation, Visualization, Methodology, Writing – review and editing; Karin Schumacher, Conceptualization, Formal analysis, Supervision, Funding acquisition, Methodology; Ursula Kummer, Conceptualization, Software, Formal analysis, Supervision, Funding acquisition, Validation, Investigation, Methodology, Project administration, Writing – review and editing; Klaus Harter, Conceptualization, Resources, Formal analysis, Supervision, Funding acquisition, Validation, Investigation, Methodology, Writing – original draft, Project administration, Writing – review and editing

### Author ORCIDs

Leander Rohr  http://orcid.org/0000-0003-4592-4197
Karin Schumacher  http://orcid.org/0000-0001-6484-8105
Klaus Harter  http://orcid.org/0000-0002-2150-6970

### Decision letter and Author response

Decision letter https://doi.org/10.7554/eLife.73031.sa1
Author response https://doi.org/10.7554/eLife.73031.sa2

## Additional files

### Supplementary files

- Appendix 1—figure 1—source data 1. Raw data underlying the representation of the results.
- Appendix 1—figure 4—source data 1. Raw data underlying the representation of the results.
- Appendix 1—figure 5—source data 1. Raw data underlying the representation of the results.

- Appendix 1—figure 6—source data 1. Raw data underlying the representation of the results.
- Transparent reporting form

## Data availability

All data generated and analysed during this study are included in the manuscript and Appendix 1. Raw and metadata are provided for Figures 4, 5, 6, 7 and 8 as well as for Appendix 1 Figures 2, 3, 4 and 6. Figure 1 represents scheme of early BRI1 signaling and Figure 2 the scheme of the used model structure. Predominantly published scRNA-Seq data were used for Figure 3. Modelling codes are available in Appendix 1 - model information.

The following previously published dataset was used:

| Author(s) | Year | Dataset title | Dataset URL | Database and Identifier |
|---|---|---|---|---|
| Denyer T, Ma X, Klesen S, Scacchi E, Nieselt K, Timmermans MC | 2019 | Spatiotemporal development trajectories in the Arabidopsis root revealed using high-throughput single-cell RNA sequencing | https://www.ncbi.nlm.nih.gov/geo/query/acc.cgi?acc=GSE123818 | NCBI Gene Expression Omnibus, GSE123818 |

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

# Appendix 1

## Model information

Throughout this section we will indicate to concentrations using [], e.g. [$BRI1$]. Proteins or ions, which can appear in different compartments will have the respective compartment indicated in the subscript, e.g. [$H_{out}^+$]. Compartment volumes will be indicated by $V$, surface areas by $A$, with the compartment name indicated in the subscript. Time-dependent volumes or global quantities are indicated by (t), e.g. *cell wall thickness(t)*.

## Compartments

The computational model was set up to describe the behavior of one epidermis cell in the *Arabidopsis thaliana* root. Initially, the compartment sizes and concentrations of model species were set such that the model describes an epidermis cell in the early elongation zone (EZ). All unknown parameters were estimated based on the pathway's behavior in this root zone. To verify the model behavior we changed the setting to now describe an epidermis cell in the meristematic zone (MZ) and predicted the time-course and dose-response behavior.

The compartments were defined so that the cytosol, membrane area and vacuole (as well as the vacuolar area) are fixed. The cell wall volume on the other hand is defined as the product of the membrane area and the cell wall thickness. The (initial) sizes for all compartments are listed in *Appendix 1—table 1*.

As one of the responses to BR signaling is the swelling of the cell wall, both cell wall thickness and volume are time-dependent and change according to the acidification of the cell wall:

$$V_{cell\ wall}(t) = A_{cell\ surface} \cdot cell\ wall\ thickness(t)$$

The cell wall thickness itself is calculated by an ODE using the current value and the instability caused by the acidification: $\frac{d(cell\ wall\ thickness)}{dt} = scaling\ factor \cdot cell\ wall\ instability(t) \cdot cell\ wall\ thickness(t) \cdot stimulation$

The cell wall instability is calculated based on the current level of acidification and limited by how close the cell wall thickness is to the maximally allowed value:

$$cell\ wall\ instability(t) = stimulation \cdot \frac{1}{1+e^{-0.001\left([H_{out}^+]-1.2\cdot proton\ readout\right)}} \cdot \left(1 - \frac{1}{1+e^{-10^7 \cdot \left(Cell\ wall\ thickness(t)-5.0\cdot10^{-6}\right)}}\right)$$

Altogether this allows the model to capture not only the pH and Em change after BR stimulation but also the early cell morphological change of cell wall swelling in preparation of cell elongation.

**Appendix 1—table 1.** Overview of model compartments and (initial) sizes for both MZ and early EZ. [a] calculated by multiplying the membrane area with the cell wall thickness (***Wilma van Esse et al., 2011***; ***Caesar et al., 2011***). [b] estimated volume based on cell dimensions and cellular volume (***Wilma van Esse et al., 2011***). [c] estimated surface area, included as scaling factor in the global quantities.

| Root zone | Compartment | Size |
| --- | --- | --- |
| Meristematic zone | cytosol | $8.47 \times 10^{-13}$ dm³ |
| | membrane | $7.67 \times 10^{-8}$ dm² |
| | cell wall[a] | $3.03 \times 10^{-13}$ dm³ |
| | vacuole | NA |
| | vacuolar surface | NA |
| Early elongation zone | cytosol | $2.271 \times 10^{-12}$ dm³ |
| | membrane | $2.098 \times 10^{-7}$ dm² |
| | cell wall[a] | $8.2871 \times 10^{-13}$ dm³ |

*Appendix 1—table 1 Continued on next page*

*Appendix 1—table 1 Continued*

| Root zone | Compartment | Size |
|---|---|---|
| | vacuole[b] | $2.352 \times 10^{-12}$ dm$^3$ |
| | vacuolar surface[c] | $1.087 \times 10^{-7}$ dm$^2$ |

## Overview of model components

**Appendix 1—table 2.** Protein are specified by the Uniprot identifier (***Bairoch et al., 2005***) and the corresponding gene ID.

For ions and chemical compounds, the ChEBI (Chemical Entities of Biological Interest ***Degtyarenko et al., 2008***) identifier is used instead. The initial concentrations of all un-phosphorylated species and complexes between proteins were set to 0 pM.

| Species | Uniprot ID / ChEBI ID | Gene ID | Initial Concentration | Source |
|---|---|---|---|---|
| BRI1 | O22476 | At4g39400 | 0.182 633 pM | ***Wilma van Esse et al., 2011*** |
| BAK1 | Q94F62 | At4g33430 | 0.099 632 pM | ***Wilma van Esse et al., 2011*** |
| BIR3 | | | 0.237 423 11 pM | this study |
| AHA* | | | 0.232 442 pM | $[AHA1] + [AHA2]$ |
| AHA1 | P20649 | At2g18960 | 0.116 221 pM | assumption: $\frac{AHA1}{AHA2} \approx \frac{1}{1}$ |
| | | | | mRNA data (eFP Browser) |
| | | | | ***Winter et al., 2007*** |
| AHA2 | P19456 | At4g30190 | 0.116 221 pM | this study |
| AHA CT* | C-terminus of AHAs | | 0.232 442 pM | assumed to be $[AHA1] + [AHA2]$ |
| BKI1 | Q9FMZ0 | At5g42750 | 0.219 16 pM | assumption: $1.2 \cdot [BRI1]_{t=0}$ |
| BIK1 | O48814 | At2g39660 | 0.219 16 pM | assumption: $1.2 \cdot [BRI1]_{t=0}$ |
| CNGC10 | Q9LNJ0 | At1g01340 | 0.1 pM | |
| H$^+_{in}$ | 24636 | - | 63 000 pM | |
| H$^+$out | 24636 | - | fitted to data | |
| K$^+$out | 29103 | - | $9.8425 \times 10^9$ pM | ½ MS medium |
| K$^+_{in}$ | 29103 | - | $8.4 \times 10^{10}$ pM | ***Maathuis and Sanders, 1993*** |
| K$^+_{vac}$ | 29103 | - | $8.4 \times 10^{10}$ pM | assumed to be identical to K$^+_{in}$ |
| BL | 28277 | - | *dose* | see experimental setup |

[*]To avoid overly complicating the model we have summarized the pump activity of $AHA1$ and $AHA2$ into one reaction that is mediated by $AHA$, which is defined as the sum of $[AHA1]$ and $[AHA2]$. Similarly, regulatory function of the C-terminal regions of the AHAs is mediated by the unphosphorylated form of the C-terminus AHA CT, which represents the C-terminal regions of both $AHA1$ and $AHA2$.

## Ordinary differential equations

The differential equations of the model are composed of the individual rate laws of the biochemical and interaction reactions considered for the model (***Figure 2***). Unless otherwise indicated, reaction rates (indicated by "$k$") and affinities (indicated by "$K$") are defined locally for that particular reaction. The index of the reaction ($r_i$) and of the respective rate law ($v_i$) are identical to the numbers in the model scheme (***Figure 2***).

## Overview of reactions

**Appendix 1—table 3.** Overview of model reactions, including the reaction number (ID), the general type of rate law chosen for the respective reaction as well as available experimental parameter values.

Rate law abbreviations: MA - mass action kinetics, MM - Michaelis-Menten kinetics, CF - constant flux.

| ID | Rate law | Parameter | Value | Source |
|---|---|---|---|---|
| $r_{01}$ | modified MA | $K_i$ | approx. $7.7$ — fold for AHA2 | *Regenberg et al., 1995* |
| $r_{02}$ | modified MA | $k$ | $0.84 \times 10^{-9} \mathrm{dm\,s^{-1}}$ to $1.25 \times 10^{-9} \mathrm{dm\,s^{-1}}$ | this study |
| $r_{03}$ | modified MA | | | |
| $r_{04}$ | MA | | | |
| $r_{05}$ | MA | | | |
| $r_{06}$ | MA | | | |
| $r_{07}$ | MA | | | |
| $r_{08}$ | CF, MA | *dose* | from $0\,\mathrm{nM}$ to $1 \times 10^5 \mathrm{pM}$ | *Caesar et al., 2011* this study |
| $r_{09}$ | modified MA | $K_d$ | $7.4 \times 10^3 \mathrm{pM}$ to $5.5 \times 10^4 \mathrm{pM}$ | *Clouse, 2002* *Hohmann et al., 2018* |
| | | | | *Kinoshita et al., 2005* |
| | | | | *Wang et al., 2001* |
| | | $k_{on}$ | $9.49 \times 10^{-7} \mathrm{pMol^{-1}\,s^{-1}}$ | *Hohmann et al., 2018* |
| $r_{10}$ | modified MM | $k$ | $0.97 \mathrm{s^{-1}}$ | *Wang et al., 2014* |
| $r_{11}$ | modified MM | | | |
| $r_{12}$ | modified MA | | | |
| $r_{13}$ | MA | | | |
| $r_{14}$ | MA | | | |
| $r_{15}$ | MM | | | |
| $r_{16}$ | MM | | | |
| $r_{17}$ | MM | *time scale* | slow increase over $12\,\mathrm{h}$ | *Oh et al., 2012* |
| $r_{d1}$ | MA | | | |
| $r_{d2}$ | MA | max. $k_d$ | $1.05 \times 10^{-2} \mathrm{s^{-1}}$ | *Hohmann et al., 2018* |
| $r_{d3}$ | MA | max. $k_d$ | $1.05 \times 10^{-2} \mathrm{s^{-1}}$ | *Hohmann et al., 2018* |
| $r_{d4}$ | MA | *time scale* | residual $P_i$ after $5\,\mathrm{d}$ | *Oh et al., 2012* |
| $r_{d5}$ | MA | | | |
| $r_{d6}$ | MA | | | |
| $r_{d7}$ | MA | | | |

Reaction rate laws $v_{ID}$ for all model reactions $r_{ID}$

$r_{01}:$ $\quad v_{01} = A_{cell\,surface} \cdot k_{01} \cdot [AHA] \cdot [H_{in}^+] \cdot \frac{[AHA]}{[AHA]+Inhibition_{AHA\,CT} \cdot [AHA\,CT]}$

$r_{02}:$ $\quad v_{02} = A_{cell\,surface} \cdot k_{02} \cdot ([H_{out}^+] - [H_{in}^+])$

$r_{03}:$ $\quad v_{03} = A_{cell\,surface} \cdot (k_{03} \cdot [K_{in}^+] - k_{03} \cdot K_{eq} \cdot [K_{out}^+])$

$r_{04}:$ $\quad v_{04} = k_{04} \cdot A_{cell\,surface} \cdot [CNGC10_{open}] \cdot \frac{[K_{out}^+]}{K_d} \cdot \left(\frac{E_m}{-0.59} - 1\right)$

$r_{05}:$ $\quad v_{05} = A_{vacuole} \cdot k_{05} \cdot ([K_{in}^+] - [K_{vac}^+])$

$r_{06}:$ $\quad v_{06} = A_{cell\,surface} \cdot (k_{06} \cdot [BIR3] \cdot [BRI1] - k_{06} \cdot K_D \cdot [BIR3\,BRI1])$

$r_{07}:$ $\quad v_{07} = A_{cell\,surface} \cdot (k_{07} \cdot [BIR3] \cdot [BAK1] - k_{07} \cdot K_D \cdot [BIR3\,BAK1])$

$r_{08}:$ $\quad v_{08} = V_{cell\,wall} \cdot (k_{08} \cdot dose \cdot stimulation - k_{08} \cdot [BL])$

$r_{09}:$ $\quad v_{09} = A_{cell\,surface} \cdot (k_{on} \cdot [BL] \cdot [BRI1] - k_{off} \cdot [BRI1\,BL])$

$r_{10}:$ $\quad v_{10} = A_{cell\,surface} \cdot k_{10} \cdot [BRI1\,BL] \cdot \frac{[BKI1]}{(K_{i,BKI1}+[BKI1]) \cdot (1+\frac{[BKI1]}{K_{i,BKI1}}) \cdot (1+\frac{[BIK1]}{K_{i,BIK1}})}$

$r_{11}:$ $\quad v_{11} = A_{cell\,surface} \cdot k_{11} \cdot [BRI1\,BL] \cdot \frac{[BIK1]}{(K_{i,BIK1}+[BIK1]) \cdot (1+\frac{[BKI1]}{K_{i,BKI1}}) \cdot (1+\frac{[BIK1]}{K_{i,BIK1}})}$

$r_{12}:$ $\quad v_{12} = A_{cell\,surface} \cdot k_{12} \cdot [BRI1\,BL] \cdot \frac{[1]}{(1+\frac{[BKI1]}{K_{i,BKI1}}) \cdot (1+\frac{[BIK1]}{K_{i,BIK1}})}$ $\qquad$ (1)

$r_{13}:$ $\quad v_{13} = A_{cell\,surface} \cdot (k_{13} \cdot [BRI1p\,BL] \cdot [BAK1] - k_{off} \cdot [BAK1\,BRI1p\,BL])$

$r_{14}:$ $\quad v_{14} = A_{cell\,surface} \cdot k_{14} \cdot [BAK1\,BRI1p\,BL]$

$r_{15}:$ $\quad v_{15} = A_{cell\,surface} \cdot k_{15} \cdot [BAK1p\,BRI1pp\,BL] \cdot \frac{[AHA\,CT]}{[AHA\,CT]+K}$

$r_{16}:$ $\quad v_{16} = A_{cell\,surface} \cdot \left(k_{16} \cdot [BAK1p\,BRI1pp\,BL] \cdot \frac{[CNGC10_{closed}]}{K_M+[CNGC10_{closed}]} - k_{-16} \cdot [CNGC10_{open}]\right)$

$r_{d1}:$ $\quad v_{d1} = A_{cell\,surface} \cdot k_{off} \cdot [BRI1p\,BL]$

$r_{d2}:$ $\quad v_{d2} = A_{cell\,surface} \cdot k_{off2} \cdot [BAK1\,BRI1p\,BL]$

$r_{d3}:$ $\quad v_{d3} = A_{cell\,surface} \cdot k_{off2} \cdot [BAK1p\,BRI1pp\,BL]$

$r_{d4}:$ $\quad v_{d4} = A_{cell\,surface} \cdot k_{off3} \cdot [BAK1p\,BRI1inact\,BL]$

$r_{d5}:$ $\quad v_{d5} = A_{cell\,surface} \cdot k_{d5} \cdot [BKI1pY211]$

$r_{d6}:$ $\quad v_{d6} = A_{cell\,surface} \cdot k_{d6} \cdot [BIK1p]$

$r_{d7}:$ $\quad v_{d7} = A_{cell\,surface} \cdot k_{d7} \cdot [AHA\,CTp]$

Model ODEs

$$\frac{d([BRI1] \cdot A_{cell\ surface})}{dt} = -v_{09} + v_{d1} + v_{d2} + v_{d3} + v_{d4}$$

$$\frac{d([BL] \cdot A_{cell\ surface})}{dt} = -v_{09} + v_{d1} + v_{d2} + v_{d3} + v_{d4} + v_{08}$$

$$\frac{d([BKI1pY211] \cdot V_{cytosol})}{dt} = +v_{10} - v_{d5}$$

$$\frac{d([BKI1*] \cdot A_{cell\ surface})}{dt} = -v_{10} + v_{d5}$$

$$\frac{d([AHA\ CTp] \cdot A_{cell\ surface})}{dt} = +v_{15} - v_{d7}$$

$$\frac{d([AHA\ CT] \cdot A_{cell\ surface})}{dt} = -v_{15} + v_{d7}$$

$$\frac{d([BAK1] \cdot A_{cell\ surface})}{dt} = -v_{07} - v_{13} + v_{d2} + v_{d3} + v_{d4}$$

$$\frac{d([BAK1\ BRI1p\ BL] \cdot A_{cell\ surface})}{dt} = +v_{13} - v_{d2} - v_{14}$$

$$\frac{d([BAK1p\ BRI1pp\ BL] \cdot A_{cell\ surface})}{dt} = +v_{14} - v_{17} - v_{d3}$$

$$\frac{d([BAK1p\ BRI1inact\ BL] \cdot A_{cell\ surface})}{dt} = +v_{17} - v_{d4}$$

$$\frac{d([BIK1] \cdot A_{cell\ surface})}{dt} = -v_{11} + v_{d6}$$

$$\frac{d([BIK1p] \cdot A_{cell\ surface})}{dt} = +v_{11} - v_{d6}$$

$$\frac{d([BRI1\ BL] \cdot A_{cell\ surface})}{dt} = +v_{09} - v_{12}$$

$$\frac{d([BRI1p\ BL] \cdot A_{cell\ surface})}{dt} = +v_{12} - v_{13} - v_{d1}$$

$$\frac{d([BIR3] \cdot A_{cell\ surface})}{dt} = -v_{07} - v_{06}$$

$$\frac{d([BIR3\ BAK1] \cdot A_{cell\ surface})}{dt} = +v_{07}$$

$$\frac{d([BIR3\ BRI1] \cdot A_{cell\ surface})}{dt} = +v_{06}$$

$$\frac{d([H^+_{out}] \cdot V_{cell\ wall})}{dt} = +v_{01} - v_{02}$$

$$\frac{d([CNGC10_{open}] \cdot A_{cell\ surface})}{dt} = +v_{16}$$

$$\frac{d([CNGC10_{closed}] \cdot A_{cell\ surface})}{dt} = +v_{16}$$

$$\frac{d([K^+_{in}] \cdot V_{cell})}{dt} = +v_{04} + v_{03} - v_{05}$$

$$\frac{d([K^+_{vac}] \cdot V_{vacuole})}{dt} = +v_{05}$$

(2)

## Global quantities

The computational model also comprises a number of global quantities that are important for the model setup and analysis. First, the net change in charge distribution and in the membrane potential are calculated as global quantities. The net change in charge distribution is calculated based on the change in intracellular potassium amount ($K^+_{in}$) and in the extracellular proton amount ($H^+_{out}$), the Faraday constant $F$ and a scaling factor from *pmol* to *mol*:

$$\Delta Q(t) = (([K^+_{in}] - [K^+_{in,t=0}]) \cdot V_{cell} - ([H^+_{out}] - [H^+_{out,0}]) \cdot V_{cell\ wall(t)}) \cdot F \cdot factor_{pmol\ to\ mol}$$
$$with : F = 96485.33212\,C\,mol^{-1}$$

The corresponding membrane potential change $\Delta E_m$ is then calculated based on the change in charge distribution $\Delta Q$, the specific membrane capacitance and the membrane area.

$$\triangle E_m(t) = \frac{net\ charge\ distribution\ change\ \triangle Q(t)}{specific\ capacitance * membrane\ area}$$

The membrane potential itself as then computed based on the initial value and the calculated membrane potential change:

$$E_m(t) = E_{m,\ t=0} + \triangle E_m(t)$$

## Expression factors for BIR3 and BRI1

Factors representing the expression level of BIR3 and BRI1. 1 represents the normal expression level, 100 represents the overexpression level. These factors are used to simulate the behavior of the overexpression phenotypes.

## Global model parameters

A number of model parameters were defined as global quantities: the affinity and dissociation rate of BL from BRI1 ($K_d$ and $k_{off\,BL}$, respectively), the dissociation rate of BL from BRI1 and BAK1 ($k_{off\,BL\,BRI1\,BAK1}$), the affinity between BIR3 and BAK1 as $K_{d\,BIR3\,BAK1} = rel\,affinity\,BIR3 \dot{K}_{d\,BIR3\,BRI1}$ (with $rel\,affinity\,BIR3 < 1$), and the inhibitory constants for BIK1 and BKI1 (as $K_{i\,BKI1}$ and $K_{i\,BIK1}$, respectively).

## Events
### Stimulation
Trigger: $Model\,Time > 86400\,s$

Target: Global quantity *stimulation transient value* is set to 1 from the initial value of 0.

### pH measurements using pHusion
*A. thaliana* seedlings stably expressing SYP122-pHusion were treated with with 500 µM ortho-vanadate and the pH was measured after 30 min and 60 min based on the fluorescent ratio of mRFP and eGFP in the EZ. The measurements were conducted for n=30 seedlings. Plants treated with MS medium were taken as control, outliers were set to pH 8.

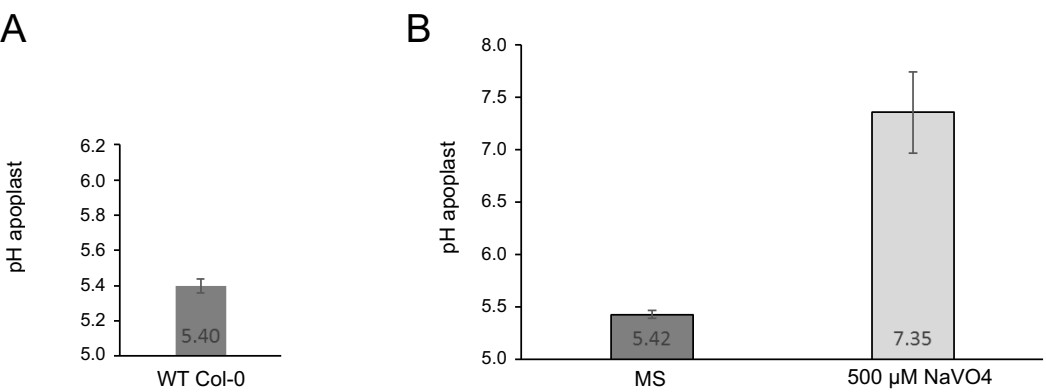

**Appendix 1—figure 1.** Measurement of the proton leak flux from the cell wall using SYP122-pHusion. (**A**) Resting pH in the EZ of the WT Col-0. Error bars represent SD (n=3). (**B**) pH after 1 h of treatment with $500\,M$ ortho-vanadate compared to control (MS). Error bars represent SD (n=30). The proton leak was estimated based on the pH difference and the average size of an epidermis cell in the mid EZ (*Wilma van Esse et al., 2011*).

The online version of this article includes the following source data for appendix 1—figure 1:

• **Appendix 1—figure 1—source data 1.** Raw data underlying the representation of the results.

## Example calculation of E$_m$ and pH change

pH 5.4 → 5.0

$$\triangle[H^+]:\quad 10^{-5.0}\,M - 10^{-5.4}\,M = 1 \cdot 10^{-5} - 3.16 \cdot 10^{-6}M = 6.019 \cdot 10^{-6}M$$

$$\triangle nH^+:\quad 6.019 \cdot 10^{-6}M \cdot 8.2892 \cdot 10^{-13}l = 4.99 \cdot 10^{-18}mol$$

$$\triangle Q:\quad 4.99 \cdot 10^{-18}mol \cdot 96485.33212\tfrac{C}{mol} = 4.81 \cdot 10^{-13}C$$

$$\triangle E_m:\quad \frac{4.81 \cdot 10^{-13}C}{0.0081 \cdot 2.098 \cdot 10^{-9}m^2} = 2.83 \cdot 10^{-2}V = 28.3mV$$

## Computational modeling enables the in silico analysis of BIR3 function
To further demonstrate the importance of modeling for the understanding of a cell physiological process, we investigated the function of the inhibitor BIR3 in the activity modulation of the nano-organized BRI1 complexes in more detail in silico. The basis for the focus on BIR3 were the observations by *Imkampe et al., 2017* regarding the activity of the BR signaling in BIR3 as well as BIR3 and BRI1 overexpressing plant in the parameter estimation and the proof of the graduated interaction of the cytplasmic domains of BIR3 with BAK1 and BRI1 (*Großeholz et al., 2020*): The pathway should be

inactive (=no acidification), when BIR3 is overexpressed, whilst the additional overexpression of BRI1 should restore the signaling activity to approximately normal levels. As shown in figure S14 A, the model was actually able to describe and represent the BR activity of the respective growth-related experimentally measured phenotypes of *Arabidopsis* plants with altered BIR3 levels (*Imkampe et al., 2017*). The accuracy of the model allowed us to investigate the behavior of different BIR3 expression levels in comparison to wild type level in the root by analyzing the pH change 20 min after stimulation with 10 nM BL. As shown in the resulting expression-response curve (Fig. S14 B), the overall response decreased with increasing concentrations of BIR3 for all model parameterizations. The model therefore suggests that it is possible for the plant to fine-tune the signaling output by adjusting the protein level of the negative regulator BIR3. Again using the model, we also analyzed the dynamics of the overall pH response at different BIR3 accumulation levels, namely in the absence of BIR3, the wild type protein amount of around 13 BIR3 molecules $\mu m^{-2}$ PM and a 10- and 100-fold overaccumulation of BIR3. Here, the actual time-course behavior of the acidification varies between the different model parameterizations as the span of possible values deviated from the average pH response for the BIR3 expression (Fig. S14 C). Depending on the parameterization, it was possible for the model to either show a strong activation that tapered off or a more gradual response over the time- frame of an hour. However, for most model parameterizations, a 10-fold overexpression of BIR3 is sufficient to inactivate the BRI1 signaling module confirming the importance of the regulation by BIR3. In summary, the modeling reveals insights into the quantitative properties of the considered cell physiological process with an accuracy that is very difficult to assess experimentally.

## Supporting figures

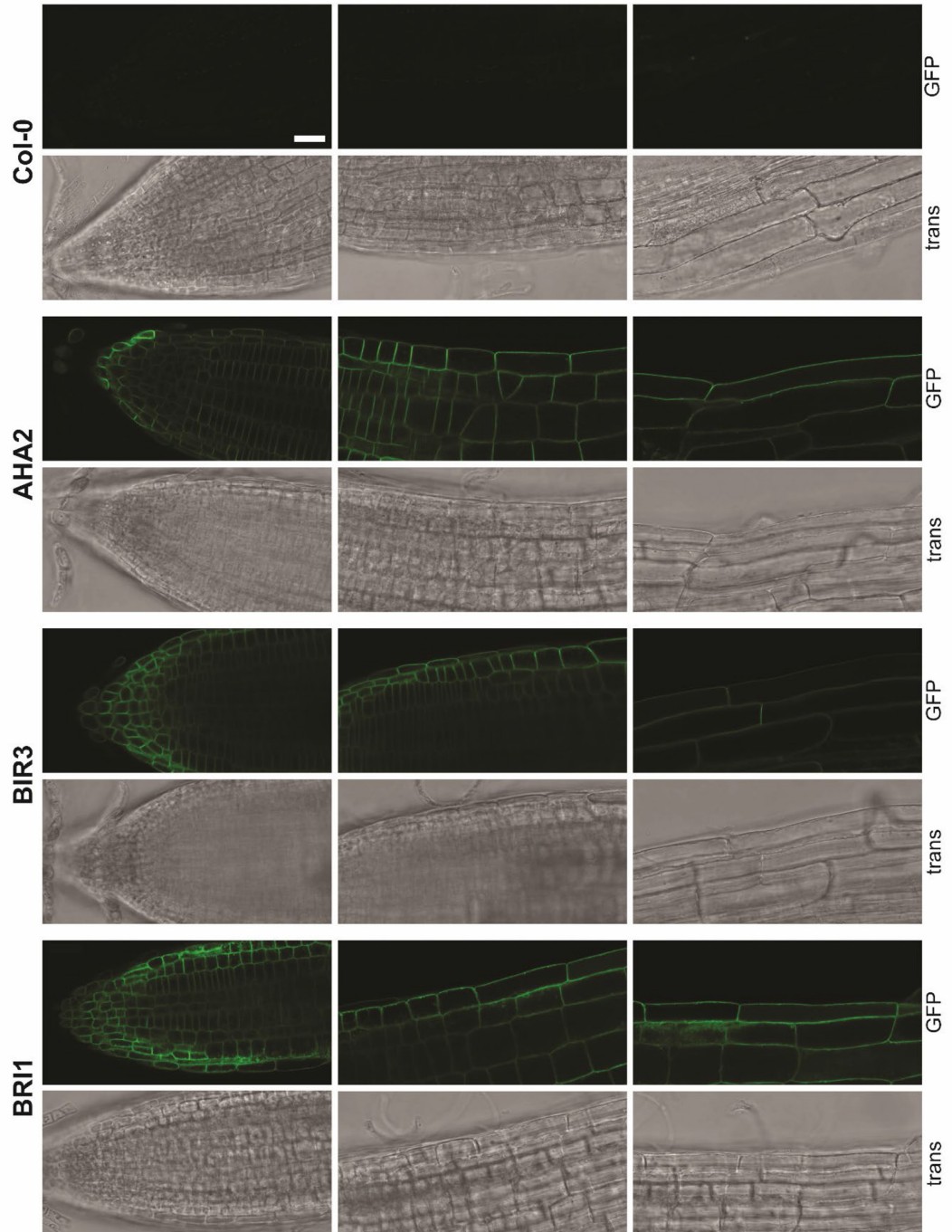

**Appendix 1—figure 2.** Exemplary images of the localization of fusions proteins. Localization along the root axis. Shown is the localization of AHA2-GFP, BIR3-GFP and BRI1-GFP, expressed under the respective native promoter in the respective mutant background (5-days-old seedlings). Col-0 (top) served as control. From top to bottom: GFP channel; transmitted light (trans). As reported before the amount of BRI1-GFP did not alter much (see *Figure 3C*; *van Esse et al., 2012*). In addition, for BIR3-GFP a homogenous fluorescence was observed, as well (see *Figure 3C*). In contrast, there was a gradient of AHA2-GFP fluorescence intensity along the root axis, being comparatively low in the meristematic zone (MZ) but high in the elongation zone (EZ) / maturation zone (see *Figure 3C*). Images were taken with a SP8 laser scanning microscope (Leica Microsystems GmbH) under the use of the HC PL APO CS2 63 x/1.20 WATER objective. For all images, the same settings were used: Argon Laser: 30%. For GFP excitation: 488 nm laser line (with adequate laser power to avoid saturation of the signal). GFP

*Appendix 1—figure 2 continued on next page*

*Appendix 1—figure 2 continued*

fluorescence was detected by an HyD detector between 500 nm – 550 nm (190 V gain, –0.01 offset). PMT Trans was used to detect transmitted light (217 V gain, offset off). By an XY-dimension of 1024x512 px and a scan speed of 200 Hz, the zoom factor was 0.75. For better visibility, the intensity values were adjusted as followed: 0–75 (AHA2) for GFP channel and 0–85 for all transmitted light channels. Scale bar represents 25 µm and applies to all partial images.

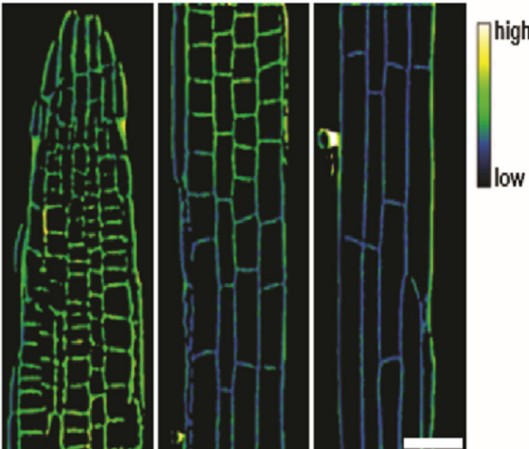

**Appendix 1—figure 3.** Representative image of the apoplastic pH of epidermal cells along the root axis of wild type *Arabidopsis* using HPTS-staining starting with the meristematic zone (MZ, left) over the transition zone (TZ, middle) to the elongation zone (EZ, right). The scale bar represents 25 µm and applies to all partial images.

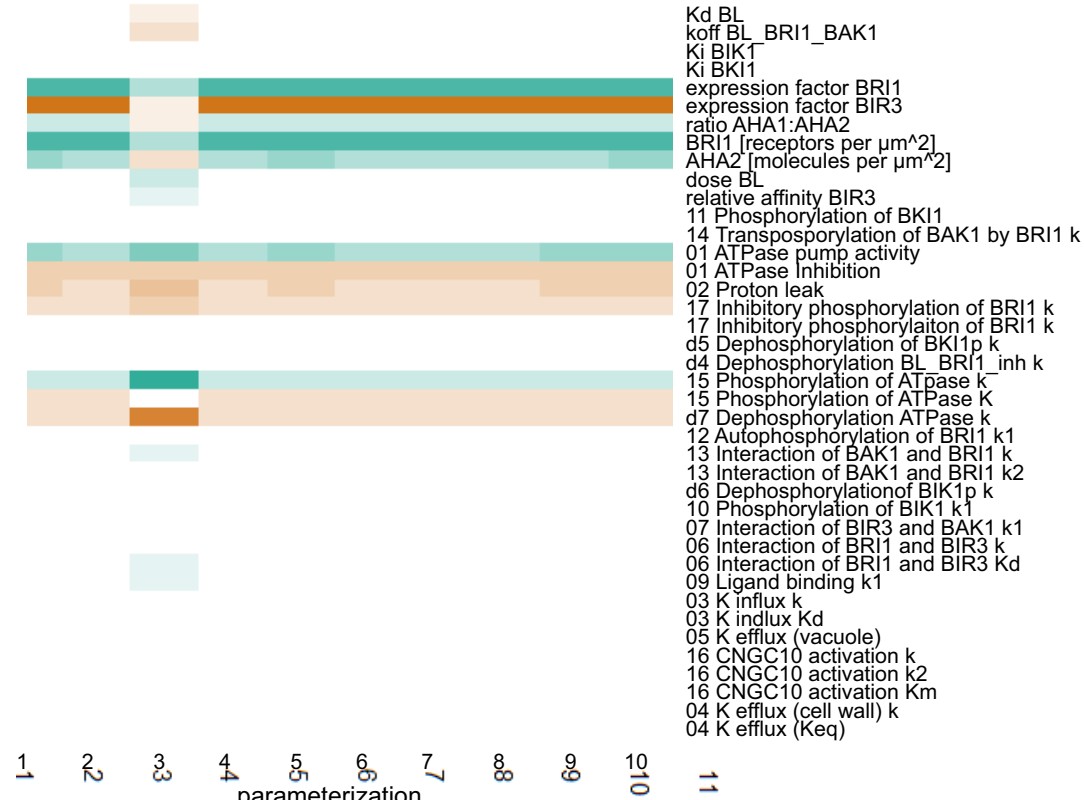

Kd BL
koff BL_BRI1_BAK1
Ki BIK1
Ki BKI1
expression factor BRI1
expression factor BIR3
ratio AHA1:AHA2
BRI1 [receptors per µm^2]
AHA2 [molecules per µm^2]
dose BL
relative affinity BIR3
11 Phosphorylation of BKI1
14 Transposporylation of BAK1 by BRI1 k
01 ATPase pump activity
01 ATPase Inhibition
02 Proton leak
17 Inhibitory phosphorylation of BRI1 k
17 Inhibitory phosphorylaiton of BRI1 k
d5 Dephosphorylation of BKI1p k
d4 Dephosphorylation BL_BRI1_inh k
15 Phosphorylation of ATPase k
15 Phosphorylation of ATPase K
d7 Dephosphorylation ATPase k
12 Autophosphorylation of BRI1 k1
13 Interaction of BAK1 and BRI1 k
13 Interaction of BAK1 and BRI1 k2
d6 Dephosphorylationof BIK1p k
10 Phosphorylation of BIK1 k1
07 Interaction of BIR3 and BAK1 k1
06 Interaction of BRI1 and BIR3 k
06 Interaction of BRI1 and BIR3 Kd
09 Ligand binding k1
03 K influx k
03 K indlux Kd
05 K efflux (vacuole)
16 CNGC10 activation k
16 CNGC10 activation k2
16 CNGC10 activation Km
04 K efflux (cell wall) k
04 K efflux (Keq)

1 2 3 4 5 6 7 8 9 10 11

parameterization

**Appendix 1—figure 4.** Scaled sensitivities of the pH change 60 min after stimulation with 10 nM BL in response to changes in the parameter and global quantities values. Color code: red - negative control, white - no influence, green - positive control. Color saturation indicates strength of the influence.

The online version of this article includes the following source data for appendix 1—figure 4:

- **Appendix 1—figure 4—source data 1.** Raw data underlying the representation of the results.

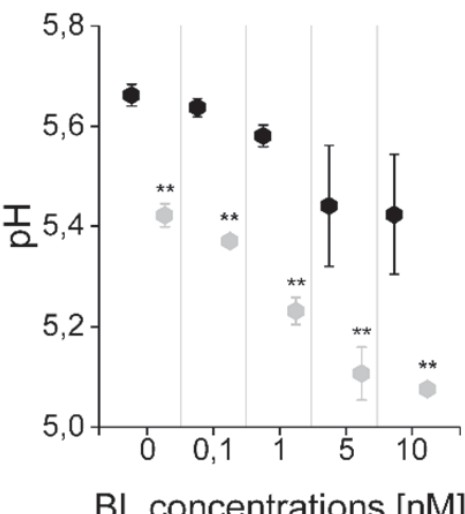

**Appendix 1—figure 5.** The computational model captures the differences in the sensitivity of apoplastic acidification between the root epidermal cells of the meristematic zone (MZ) and elongation zone (EZ) in response to BL. Black diamonds represent MZ and grey diamonds EZ. Virtual BL incubation of different concentration was
*Appendix 1—figure 5 continued on next page*

*Appendix 1—figure 5 continued*

done for 60 min. Error bars represent SD for the simulations of different model parameterizations (n=10). Statistical evaluations were performed by comparing the respective groups separately (e.g. '0 nM MZ' compared with '0 nM EZ'). For all comparisons a Two-Tailed Wilcoxon Test was applied. The black asterisks indicate statistically significant differences (**: p≤0.01). The exact p-values can be found in the attached RAW data file. The EZ cells showed a lower resting apoplastic pH and a stronger concentration-dependent response than MZ cells.

The online version of this article includes the following source data for appendix 1—figure 5:

• **Appendix 1—figure 5—source data 1.** Raw data underlying the representation of the results.

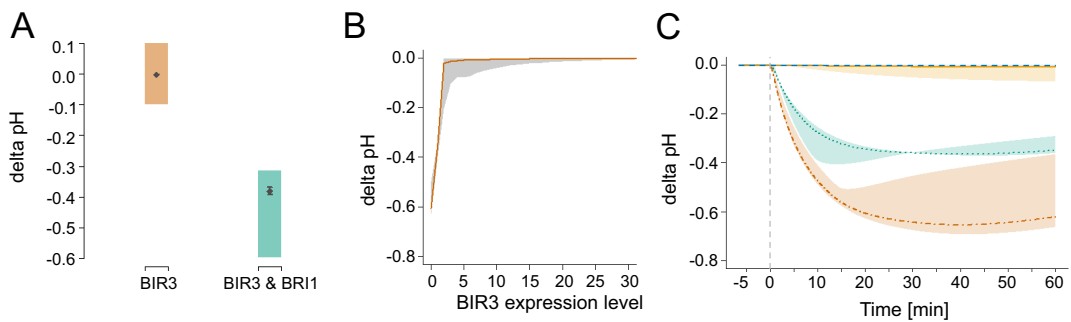

**Appendix 1—figure 6.** In silico analysis of the functional role of the negative regulator BIR3 on BL-regulated apoplastic acidification. (**A**) Modelled qualitative acidification output of plants overexpressing BIR3 and BIR3 & BRI1, respectively. The colored area represents the pH response targeted during parameter estimation, which was approximated by the activity of BR signaling indicated by the plant phenotypes (*Imkampe et al., 2017*). (**B**) BIR3-Expression-response curve. Shown is the pH change 20 min after stimulation with 10 nM BL at different BIR3 expression levels ranging from 0- (loss-of-function mutant) to 30-times the normal expression level of the wild type. The entire range of simulated responses is indicated by the shaded area, the averaged response of all models is denoted by the line. (**C**) Exemplary time-course simulations of the pH change at 0 (loss-of-function mutant, orange), 1- (wild type expression, green), 10- (yellow), and 100-fold (blue) expression of BIR3 upon virtual application of 10 nM BL. Shown is the average pH response for the respective BIR3 expression level with the span between minimal and maximal values indicated by the colored area. The virtual addition of BL at time 0 is indicated by the vertical dashed line.

The online version of this article includes the following source data for appendix 1—figure 6:

• **Appendix 1—figure 6—source data 1.** Raw data underlying the representation of the results.

