## [Editor Report]

The manuscript by Grosseholz et al. addresses a so far much overlooked aspect in plant hormone signalling systems, namely how components primarily identified by genetic means jointly regulate plant root physiology. The authors demonstrate a relevant effect of Brassinosteroid signalling on the control of root cell elongation in Arabidopsis. They used an elegant combination of (electro)physiology, computer modelling and confocal microscopy to reveal molecular candidates linking BR perception to fine-tuning of cell elongation through ion fluxes.

---

## [Decision Letter]

**Decision letter after peer review:**

Thank you for submitting your article "Computational modeling and quantitative cell physiology reveal central parameters for the brassinosteroid-regulated cell growth of the *Arabidopsis* root" for consideration by *eLife*. Your article has been reviewed by 3 peer reviewers, and the evaluation has been overseen by a Reviewing Editor and Jürgen Kleine-Vehn as the Senior Editor. The following individuals involved in review of your submission have agreed to reveal their identity: Sacco de Vries (Reviewer #1).

Essential revisions:

1) A more direct connection between fast BR responses, AHA2 gradient, and cell elongation phenotypes should be demonstrated and discussed in the revised manuscript. For that matter, reviewers suggested potential experiments that could shed more light on this particular issue and improve overall robustness of claims. Authors should try to at least address the connection between BR and AHA2 activation and if possible explore if (presumably fast) AHA2 activation impact the long-term effects on root growth and provide discussion on that matter.

2) The authors should more extensively discuss modeling details, assumptions and limitations and in particular model robustness with regard to experimental observations (some additional simulations are suggested to include in the revised version; see specific comments). Most importantly, authors should put more efforts into description of models so they are self-consistent and easy for reader to follow and reproduce results.

3) Authors should improve the quality of pH analysis and attempt to demonstrate more direct connection to CNGC10 H^+^-ATPase pump activity, i.e. is the membrane hyperpolarized upon BR treatments observed as predicted by the model? For instance, performing a localisation study of CNGC10 would help to strengthen the claims.

*Reviewer #1 (Recommendations for the authors):*

1. The paper addresses early responses such as membrane hyperpolarization, apoplast acidification and cell wall swelling. This brings me immediately to one of the unclarities of the paper, being the difference between the biochemical activation of the BRI1 receptor and is associated signaling module that is measurable in minutes and the biological effects on cell expansion (and division) on a much wider timescale of hours and days. This problem was also addressed in earlier attempts to link BR-signaling with root growth physiology using quantitative modelling. Unfortunately, this is not very clear from the present paper and because *eLife* is a general platform I think this should be introduced more explicitly in the Introduction section. Viewed in this way, it would also put the observed physiological data as done in this paper in a more meaningful context. In other words, how does the activation of the proton pump relate to the observed expansion of the cells?

2. At least in one instance a direct link between BR activated signaling, observed early effects and final downstream long term physiological effects should be shown in this paper. It should also be taken into account that at the used level of BRs (10 nM) in wild type roots growth is essentially inhibited. Therefore, it needs to be explained why application of high doses of hormone at shorter time frames (minutes) is still relevant for effects recorded at much lower concentrations at much longer (days) timeframe.

3. No use is made of prior application of BR biosynthesis inhibitors to reduce the endogenous level of ligand. For short-term experiments this is perhaps not relevant but at least a few control experiments could be included.

4. If AHA2 activity is crucial for translating the BR signal to proton pump activity and cell wall acidification this would be expected to take place in the most rapidly expanding parts of the cells in the EZ. Inspection of figure 3 does not reveal very much in terms of possible localized AHA2-GFP. I suggest examining a number of EZ cells at higher magnification for AHA2 localization, ideally in conjunction with the beautiful HPTS staining experiments in Figure 7.

5. For confirmation purposes the bri1-301 weak mutant is employed in experiments described in Figure 7. That is fine of course but it poses the problem that while the 301 roots are indeed less sensitive to BR application, their roots show little deviance from the wild type under normal growth conditions. Yet, there is indeed a clear effect on the output as measured by HPTS and MIFE. What does that mean, that the bri1-301 roots have a way to compensate? One should also caution a bit because what is measured here is the response to exogenous non-physiological levels of the hormone rather than representing the normal growth pattern of the bri1-301 roots (see also my point 2).

6. The last part of the paper describes measurements and modelling leading to the identification of a cation channel protein CNGC10. Because inclusion of this component in both the model and used as loss-of-function alleles fits precisely I think this is one of the highlights of this combined approach. While not unexpected in electrophysiological terms, a very rewarding result!

7. Finally, the attention is now turned to BIR3, an associated LRR-RLK with a reported negative effect on BR-signaling. The rationale of this twist in the story is that the authors want to use it to validate their modelling in terms of residual BR activity after increasing the dosage of BIR3. In my opinion these experiments are of interest to plant receptor specialists but deflect from the main message of the paper and can be removed without loss of clarity or included as an additional supplementary figure.

*Reviewer #2 (Recommendations for the authors):*

As indicated in the public review, the manuscript has many exciting elements. However, the relevance of AHA2 gradient is not clear nor the impact of the findings on cell elongation. Therefore, the conclusions of the authors remain partially unsupported and alternative frameworks to those proposed can be envisaged. Here below I detail more these two aspects, as well as additional recommendations:

1) There are no novel experimental evidences presented in the manuscript that directly support whether this fast response affects cell growth or cell elongation in Arabidopsis roots. Evidences on this should be provided to support many of the statements of the manuscript (e.g., in the abstract it is indicated "Thus, we established the landscape of components and parameters capable of triggering and guiding cellular elongation through the fast response to BRs, a central process in plant development"). If these evidences are not provided, statements on cell elongation should be much more mitigated, and set only as proposals/suggestions that are based on the current study and on evidences from the literature, which establish a very plausible link/hypothesis. With the current data, the functional impact of the fast BR-mediated response remains partially elusive, and hence the impact of the work is less.

2) The authors identify and nicely quantify a gradient of AHA2 protein along the root, with highest levels in the elongation zone. The authors propose that cell elongation is dependent on this gradient, such that cells in the elongation zone are more sensitive to BR (and thus elongate) than those in the meristematic zone (which thereby do not elongate). Albeit being a plausible scenario, I think there is no sufficient analysis nor data to support it, and alternative scenarios are possible:

It is not obvious to me that the experimental results in the manuscript support the relevance of AHA2 levels on BR fast response (in contrast, it is clear to me that the modeling results support it). This is because the experimental results on the pH response upon BL treatment are not compared between the meristem (MZ) and elongation zones (EZ). Instead, the results are presented in separated figures (Figure 4 for EZ and Figure 6 for MZ). When looking at both figures, the experimental data look very similar, so similar that, to my eye, with those large SE, it remains elusive whether the differences are statistically significant. In my opinion, the experimental data in Figures 4 and 6 should be put together in a single figure (please also clarify whether the experimental data from these two figures come from the same set of experiments, as they look like). The experimental data from MZ cells (Figure 6) should be compared statistically to those from EZ cells (Figure 4) to assess whether the values in meristematic cells exhibit statistically significant differences to those in the EZ, i.e. to assess whether the empirical data support that MZ cells respond different than EZ cells. As it stands now, the authors assume the experimental data exhibit these differences (they probably have checked, but do not provide the analysis). Thus, it is essential to clarify whether the empirical data support distinct responses on pH upon BL treatment of meristematic and elongating cells.

3) Next, ideally it should be assessed experimentally whether these differences between MZ and EZ cells are because of AHA2 levels, by manipulating these levels. Ultimately (and this relates to point 1) it should be also ideally addressed whether changes in cell elongation are found.

4) It has been shown previously that BR signaling through BZR and BES1 is more active in the EZ than in the MZ (Chaiwanon and Wang, Curr Biol 2015), that overexpression of the synthesis of BL drives meristems with very elongated cells (Vukasinovic et al. Nature plants 2021) and that BRI1 only at the meristem is sufficient to drive normal cell elongation (Pavelescu et al. Mol Sys Biol 2018). These results seem in contradiction with AHA2 being the limiting factor, as the authors propose (e.g. BZR response is not expected to depend on AHA2 levels; it is unclear how AHA2 without BRI1 in the EZ can drive cell elongation). How do the authors explain these published results with their current framework?

*Reviewer #3 (Recommendations for the authors):*

To strengthen the claim that AHA is rapidly activated by BL via BRI1, the activation of AHAs by BLs in the root could be tested in a time-resolved manner using the anti-pThr947 antibody, in control and bri1 mutants, to support the original findings of the Caesar et al. 2011 paper.

The representation of figures should be united to make the reading of the manuscript easier (such as figures 7CD and figure 8F and Supp.3).

The term nanocluster is used repeatedly, but the actual localization of the signaling components in nanoclusters is not tested in the paper. It might be worth imaging the nanocluster formation, or use the term nanocluster more carefully.

It is not described in the methods how the quantification of the fluorophore numbers per PM area was done. Which membranes were measured and how? It is not clear which of the epidermal cell types were used for quantification – trichoblasts or atrichoblasts? I would also appreciate example images of the gradient distribution of AHA2 along the root, not only images from the meristem.

In my opinion, standard deviation should be used to represent the error bars should be used instead of standard error of the mean.

I would strongly recommend replacing the barcharts by scatterplots or boxplots.

[Editors' note: further revisions were suggested prior to acceptance, as described below.]

Thank you for resubmitting your work entitled "Computational modeling and quantitative physiology reveal central parameters for the brassinosteroid-regulated early cell physiological processes linked to elongation growth of the *Arabidopsis* root" for further consideration by *eLife*. Your revised article has been evaluated by Jürgen Kleine-Vehn (Senior Editor) and a Reviewing Editor.

The manuscript has been improved but there are some remaining issues that need to be addressed, as outlined below:

Modelling

1) Discrepancy between model predictions and experimental data should be discussed also in terms of model limitations rather than experimental method failure. Authors should make clear that model predictions need further validation as these predictions cannot be taken for granted without experimental support.

2) There should be more critical discussion on previous work i.e. BRI1 requirement only at MZ (Pavelescu et al. Mol Sys Biol 2018) is not discussed.

3) Predictions presented in figure 7D would benefit from the comparison of pH measurements in roots. If this is not feasible authors should discuss why this would be the case

4) Equation for "cell wall thickness" in line 895 seems incorrect in the description, please recheck. Term Cell wall instability" should be made clearer.

Experimental

1) Link between BR and AHA: Manuscript by Li et al. 2022 in Plant Phys (https://academic.oup.com/plphys/advance-article/doi/10.1093/plphys/kiac194/6576642) should be discussed in the revised version as it carried data relevant to conclusions provided in this manuscript.

2) Appendix 1 figure 2 (counts of AHA) would benefit from higher quality images and quantifications in order to convince Reviewers. In particular, roots look stressed and there are many dead cells which would likely impact on provided quantifications.

3) We strongly encourage authors to perform short term BR treatments (1-10nM BL) and measure root growth.

4) The authors should discuss the predicted absence of CNGC10 from the root epidermis in the text.

*Reviewer #2 (Recommendations for the authors):*

I appreciate the authors' efforts to address the criticisms. I believe the results are of interest. The model is better defined now and the flow of the manuscript has improved. There are more comparisons between simulations and experiments, as requested. Yet, the authors have not addressed entirely satisfactorily in my opinion the concerns raised. Here below I detail those aspects that remain unclear to me and which I suggest taking into account:

1) (Related to Ad 2 # reviewer 2) My first issue comes from the conclusions they raise based on their experimental data from Figures 4 and 6, i.e. on the comparison between the response to BL of EZ and MZ cells. The authors have fully addressed my issue regarding this comparison: they have statistically assessed whether there are differences between MZ and EZ cells. They find that experimental data do not confirm a difference between MZ and EZ cells in pH value nor on its response to BL, as opposed to the model, which predicts that there is a difference. Specifically, the authors now state, lines 287-292: "However, due to the limitation in the sensitivity of the HPTS method and the biological variability in the different root preparations, the difference in the BL-induced acidification responses between MZ and EZ epidermal cells could not be statistically established. The modeling approach is therefore of great advantage for the detection and characterization of small, experimentally hardly comprehensible, cell physiological response differences (Appendix 1Figure 5)." I do not agree entirely with the last sentence. In my opinion there are, at least, two different options: Option 1) The modeling approach shows small differences which experimentally are difficult to be observed, as the authors claim in their last sentence; Option 2) Plants really do not have the differences predicted by the model. Hence, the model prediction (and hence the model) is not correct. I think Option 2 should not be omitted and hence I propose the last sentence is modified such that emphasis is made on the prediction that the model makes as opposed to the emphasis on assuming that the model is correct (and hence detects small differences).

2) (Related to AD. 4 from # reviewer 2) BRI1 requirement only at MZ (Pavelescu et al. Mol Sys Biol 2018) is not discussed.

3) (Related to AD. 6 from # reviewer 2) In figure 7D the authors present simulation results for pH response under BL treatment for the WT and for bri301 in order to confirm the experimental data in Figure 7A-C. However, in 7A-C no pH measurements are made. Instead, the ion fluxes are measured. Can the simulation data be related more tightly to the experimental data? E.g. provide the ion fluxes, besides pH? Alternatively, measure pH in the plants?

4) (Related to AD. 8 and AD.10 from # reviewer 2) It still remains unclear how "cell wall instability" and "cell wall thickness" are computed. I think Equation for "cell wall thickness" in line 895 is incorrect (I think it is correctly computed in the manuscript but wrongly written down in that line). Besides, what is "Cell wall(t)" in the Eq. for "Cell wall instability" (line 896)?

*Reviewer #3 (Recommendations for the authors):*

In the re-submitted version of the manuscript and the response letter, the authors have solved most of my comments by discussing my concerns or by refusing them as being out of scope of the manuscript.

I would like to point out several of these comments and the author's response:

1) Strengthening the link between BR signaling and AHA activation:

The authors argue that in the Lin et al., 2015 paper, phosphorylation of AHA at the residue Ser 315 and Thr 328 was demonstrated. It is, however, not clear whether this particular phosphorylation leads to AHA activation. Moreover, the Lin paper uses suspension cell culture, and this phosphorylation is apparent in the first 5 minutes, and in 3h of treatment it diminishes. Therefore my critique from the previous version remains – the central component of the model (activation of AHAs by BL signaling) should be experimentally verified.

I understand that this is a difficult task. Possibly a proxy of the AHA activation could be used instead. For example, a simple root growth rate measurement experiment where BL would be added during the timecourse could help strengthen the rapid effect of BL on AHA activation, one would expect a similar effect as in case of fussicoccin – if BL activates AHAs, this should lead at least to a transient increase in growth rate. Such a simple readout of the AHA activity could be then used for wild type roots as well as the bri1-301 and cngc10 mutant.

2) Localization of CNCG10 expression

I understand that cloning the CNGC10 promoter and translational fusions might be very laborious or out of scope of this manuscript. However, the authors claim in the text that the CNGC10 is expressed in all cell types of the root (according to eFP browser), and in the response to reviewers claim that CNGC10 is induced by protoplasting, and therefore the single cell analysis cannot be performed. However, the data in the eFP browser partially originate from protoplasted cells of the root, and in the eFP browser (http://bar.utoronto.ca/efp/cgi-bin/efpWeb.cgi ), I do not see protoplasting alert. Instead, CNGC10 seems to express predominantly in inner tissues of the root and is absent from epidermis (according to the 'root' and 'single cell' options). These expression data are only a proxy of the actual protein presence, however, I think the authors should discuss the predicted absence of CNGC10 from the root epidermis in the text.

3) Quantification of the AHA molecules along the longitudinal root axis.

In the previous review, I asked the authors for more details about AHA quantification and requested some sample images of the AHA gradient. In the current version, the Appendix 1 Figure 2 shows several roots which look very stressed and show signs of dead cells in the meristematic zone, and have a short meristem and elongation zones. These roots do not look like healthy 7-d old roots. The example picture of AHA2-GFP does not show a continuous epidermal cell file on which the number of AHAs could be quantified. The figure legend states these are 7d old roots, while in the methods, the authors state that 5d old roots were used. I am worried that if such images were used for the quantification of AHA molecule numbers, the data might be noisy due to the status of these roots. Moreover, I asked for a more detailed description of the method of molecule number quantification, the authors responded that now the method is described in the methods under "confocal imaging", but I could not find any description of the method in this section. Even though this manuscript is mainly a computation one, I think the authors should clearly explain and demonstrate that the experimental inputs are solid.

In summary, as I am not a modeler, I do not feel qualified to judge the quality of the modelling part, and so I focused on the experimental part of the manuscript. Unfortunately, from my (experimental) point of view, the experimental data that serve as input data for modelling are not solid enough.

On the other hand, I find the focus of the paper as well as the model predictions important and interesting. I hoped my concerns would be addressed during the revision process, but my concerns remain similar as in the case of the previous version of the manuscript.

---

## [Author Response]

Essential revisions:1) A more direct connection between fast BR responses, AHA2 gradient, and cell elongation phenotypes should be demonstrated and discussed in the revised manuscript. For that matter, reviewers suggested potential experiments that could shed more light on this particular issue and improve overall robustness of claims. Authors should try to at least address the connection between BR and AHA2 activation and if possible explore if (presumably fast) AHA2 activation impact the long-term effects on root growth and provide discussion on that matter.

In the introduction, we now go into the connection between fast BR responses, AHA2 activity and cell elongation phenotypes in more detail and have consulted additional literature for this purpose. In this context, we would particularly like to refer to the publications (Caesar et al., 2011; Witthöft et al., 2011). In these, the link between BR perception and AHA2 activity was described in detail.

The long-term effects of BR-caused AHA2 activation on long-term effects on root growth are now discussed in more detail in the Discussion section.

2) The authors should more extensively discuss modeling details, assumptions and limitations and in particular model robustness with regard to experimental observations (some additional simulations are suggested to include in the revised version; see specific comments). Most importantly, authors should put more efforts into description of models so they are self-consistent and easy for reader to follow and reproduce results.

We now describe the model in much more detail so that we feel that it is easier for the reader to follow and to understand the results. To check the better readability of the manuscript, we gave it to experimentally working, unbiased colleagues with no link to the project for proof reading and incorporated their comments. As suggested by reviewer 2, we also included the model´s prediction for the apoplastic acidification response of the *bri1-301* mutant compared to wild type (see Figure 7 E).

3) Authors should improve the quality of pH analysis and attempt to demonstrate more direct connection to CNGC10 H^+^-ATPase pump activity, i.e. is the membrane hyperpolarized upon BR treatments observed as predicted by the model? For instance, performing a localisation study of CNGC10 would help to strengthen the claims.

After my request, the Reviewing Editor, Dr. Krzysztof Wabnik, and the Senior Editor, Dr. Jürgen Kleine-Vehn, have waived the proof of the connection of AHA pump and CNGC10 transport activity due to the extensive, hardly feasible experimental effort (tissue-specific single cell analysis of the membrane potential). The requested localization study of CNGC10 was already part of the first submission: The transporter (CNGC10-GFP) is found in the PM and co-localizes with AHA2-RFP, BAK1-mCherry and BRI1-RFP (Figure 8 D).

Reviewer #1 (Recommendations for the authors):1. The paper addresses early responses such as membrane hyperpolarization, apoplast acidification and cell wall swelling. This brings me immediately to one of the unclarities of the paper, being the difference between the biochemical activation of the BRI1 receptor and is associated signaling module that is measurable in minutes and the biological effects on cell expansion (and division) on a much wider timescale of hours and days. This problem was also addressed in earlier attempts to link BR-signaling with root growth physiology using quantitative modelling. Unfortunately, this is not very clear from the present paper and because eLife is a general platform I think this should be introduced more explicitly in the Introduction section. Viewed in this way, it would also put the observed physiological data as done in this paper in a more meaningful context. In other words, how does the activation of the proton pump relate to the observed expansion of the cells?

We have now gone into detail in the introduction about the connection between proton pumping activity and cell elongation.

2. At least in one instance a direct link between BR activated signaling, observed early effects and final downstream long term physiological effects should be shown in this paper. It should also be taken into account that at the used level of BRs (10 nM) in wild type roots growth is essentially inhibited. Therefore, it needs to be explained why application of high doses of hormone at shorter time frames (minutes) is still relevant for effects recorded at much lower concentrations at much longer (days) timeframe.

The link between BR activated signaling, proton pump activity and long-term physiological effects are given especially in the paper of Pacifici et al. (2018). However, this manuscript does not provide enough quantitative data with sufficient cellular resolution (e.g. AHA2 expression level along the root axis, apoplastic acidification) on which the establishment of a model could be based.

Possible explanations of the concentration problem are now discussed in Discussion.

3. No use is made of prior application of BR biosynthesis inhibitors to reduce the endogenous level of ligand. For short-term experiments this is perhaps not relevant but at least a few control experiments could be included.

BR biosynthesis inhibitors such as brassinazole are needed to be pre-incubated in the tissue for an extended time before the actual experiment starts. Since an indirect interference of this pre-incubation with the fast response system, we are investigating, cannot be excluded, we would like to dispense with these experiments. Nicely, the computational model is able to predict the growth behavior of BR biosynthetic mutant cells: The cells do not grow at all independent where they are placed along the root axis. In the end, this leads to a short root phenotype, as it is observed in BR biosynthetic mutants. In order not to increase the volume of data even further, we would like to dispense with the presentation of these results. They will be part of a follow-up manuscript.

4. If AHA2 activity is crucial for translating the BR signal to proton pump activity and cell wall acidification this would be expected to take place in the most rapidly expanding parts of the cells in the EZ. Inspection of figure 3 does not reveal very much in terms of possible localized AHA2-GFP. I suggest examining a number of EZ cells at higher magnification for AHA2 localization, ideally in conjunction with the beautiful HPTS staining experiments in Figure 7.

To reduce the size of figure 3, we have transferred the cell biological data for the GFP fusion proteins to Appendix 1 Figure 4. There, the localisation of AHA2-GFP is also shown in more detail in EZ cells.

Since the pH gradient along the root axis measured with HPTS has been published several times, we also have transferred our own confirming measurements to the Suppl. Data (formerly Figure 7 B, now Appendix 1 Figure 5). This reduces the complexity of figure 7.

5. For confirmation purposes the bri1-301 weak mutant is employed in experiments described in Figure 7. That is fine of course but it poses the problem that while the 301 roots are indeed less sensitive to BR application, their roots show little deviance from the wild type under normal growth conditions. Yet, there is indeed a clear effect on the output as measured by HPTS and MIFE. What does that mean, that the bri1-301 roots have a way to compensate? One should also caution a bit because what is measured here is the response to exogenous non-physiological levels of the hormone rather than representing the normal growth pattern of the bri1-301 roots (see also my point 2).

The mutant has only a weakly expressed root phenotype at the corresponding temperature. This allows the comparative MIFE and HPTS measurements of the mutant and the wild type, as the root morphologies are hardly different. However, the lower activity of the fast BR response mechanism is measurable in the *bri1-301* mutant *via* both MIFE and HPTS and is also predicted by our model (see new Figure 7D). It cannot be excluded that for instance feedback loops at gene expression level or long-term modification of the activity of other growth-promoting hormones are responsible for such a compensation. What the molecular mechanism of such compensation might look like must be shown by future experiments, which, however, is far beyond the scope of the present manuscript.

The complexity of long-term compensatory molecular mechanisms is poorly understood molecularly (no data) and difficult to capture in quantitative modelling. In fact, these poorly understood parameters and constraints have been one reason we have focused on the fast response pathway and its constituent elements. However, this leads to the problem that we cannot fully quantitatively link apoplastic acidification, alteration of the membrane potential and wall swelling to long-term elongation growth.

6. The last part of the paper describes measurements and modelling leading to the identification of a cation channel protein CNGC10. Because inclusion of this component in both the model and used as loss-of-function alleles fits precisely I think this is one of the highlights of this combined approach. While not unexpected in electrophysiological terms, a very rewarding result!

Thank you for this positive statement. The prediction of missing elements or the quantitative “playing around” with the amount or activity of constituent elements was and is one of the rational to do and extend modelling approaches in future [see also “Ad 7)/Figure 9]. Computational modelling will in general help us to understand the complexity of a given perception-response system in much more detail.

7. Finally, the attention is now turned to BIR3, an associated LRR-RLK with a reported negative effect on BR-signaling. The rationale of this twist in the story is that the authors want to use it to validate their modelling in terms of residual BR activity after increasing the dosage of BIR3. In my opinion these experiments are of interest to plant receptor specialists but deflect from the main message of the paper and can be removed without loss of clarity or included as an additional supplementary figure.

We transferred figure 9 to the Appendix (Appendix 1 Figure 2).

Reviewer #2 (Recommendations for the authors):As indicated in the public review, the manuscript has many exciting elements. However, the relevance of AHA2 gradient is not clear nor the impact of the findings on cell elongation. Therefore, the conclusions of the authors remain partially unsupported and alternative frameworks to those proposed can be envisaged. Here below I detail more these two aspects, as well as additional recommendations:1) There are no novel experimental evidences presented in the manuscript that directly support whether this fast response affects cell growth or cell elongation in Arabidopsis roots. Evidences on this should be provided to support many of the statements of the manuscript (e.g., in the abstract it is indicated "Thus, we established the landscape of components and parameters capable of triggering and guiding cellular elongation through the fast response to BRs, a central process in plant development"). If these evidences are not provided, statements on cell elongation should be much more mitigated, and set only as proposals/suggestions that are based on the current study and on evidences from the literature, which establish a very plausible link/hypothesis. With the current data, the functional impact of the fast BR-mediated response remains partially elusive, and hence the impact of the work is less.

Throughout the manuscript, the relevant formulations were toned down and presented as "link" or "hypothesis".

2) The authors identify and nicely quantify a gradient of AHA2 protein along the root, with highest levels in the elongation zone. The authors propose that cell elongation is dependent on this gradient, such that cells in the elongation zone are more sensitive to BR (and thus elongate) than those in the meristematic zone (which thereby do not elongate). Albeit being a plausible scenario, I think there is no sufficient analysis nor data to support it, and alternative scenarios are possible:It is not obvious to me that the experimental results in the manuscript support the relevance of AHA2 levels on BR fast response (in contrast, it is clear to me that the modeling results support it). This is because the experimental results on the pH response upon BL treatment are not compared between the meristem (MZ) and elongation zones (EZ). Instead, the results are presented in separated figures (Figure 4 for EZ and Figure 6 for MZ). When looking at both figures, the experimental data look very similar, so similar that, to my eye, with those large SE, it remains elusive whether the differences are statistically significant. In my opinion, the experimental data in Figures 4 and 6 should be put together in a single figure (please also clarify whether the experimental data from these two figures come from the same set of experiments, as they look like). The experimental data from MZ cells (Figure 6) should be compared statistically to those from EZ cells (Figure 4) to assess whether the values in meristematic cells exhibit statistically significant differences to those in the EZ, i.e. to assess whether the empirical data support that MZ cells respond different than EZ cells. As it stands now, the authors assume the experimental data exhibit these differences (they probably have checked, but do not provide the analysis). Thus, it is essential to clarify whether the empirical data support distinct responses on pH upon BL treatment of meristematic and elongating cells.

As mentioned above, the manuscript is designed in a way that the mutual interplay between experiment and modelling becomes visible. For this reason, we would continue to leave the EZ and MZ acidification data (dose-response, kinetics) in separate Figure 4 and Figure 6. The modelling (Figure 5) is based on the results of the EZ (Figure 4) and were then used to model the MZ, whose predictions were in turn verified by the MZ experiments (Figure 6).

We made the comparison of the BL-induced acidification response between MZ and EZ cells and a statistical difference could not be established. To our opinion, this is due to the limitation in the sensitivity of the HPTS method and the biological variability in the different root preparations (e.g. the overall physiological status of the seedlings at the day of HPTS analysis). The modelling approach is therefore of great advantage for the detection and characterization of small, experimentally hardly comprehensive but crucial response differences (see Appendix 1 Figure 6). This issue is also addressed in the Results section of the revised manuscript.

3) Next, ideally it should be assessed experimentally whether these differences between MZ and EZ cells are because of AHA2 levels, by manipulating these levels. Ultimately (and this relates to point 1) it should be also ideally addressed whether changes in cell elongation are found.

Of course, it would be nice to have such data. However, although we are working hard on this, we are currently not able to inducible down- or up-regulate the AHA2 amount in spatiotemporal and a cell type-specific resolution along the root axis during root development. According to Pacifici et al. (2018), the inducible, cross-tissue up-regulation of AHA2 results in the enhancement of cell length in the MZ. We discuss this in the text.

4) It has been shown previously that BR signaling through BZR and BES1 is more active in the EZ than in the MZ (Chaiwanon and Wang, Curr Biol 2015), that overexpression of the synthesis of BL drives meristems with very elongated cells (Vukasinovic et al. Nature plants 2021) and that BRI1 only at the meristem is sufficient to drive normal cell elongation (Pavelescu et al. Mol Sys Biol 2018). These results seem in contradiction with AHA2 being the limiting factor, as the authors propose (e.g. BZR response is not expected to depend on AHA2 levels; it is unclear how AHA2 without BRI1 in the EZ can drive cell elongation). How do the authors explain these published results with their current framework?

These topics are now discussed in the Discussion section

Reviewer #3 (Recommendations for the authors):To strengthen the claim that AHA is rapidly activated by BL via BRI1, the activation of AHAs by BLs in the root could be tested in a time-resolved manner using the anti-pThr947 antibody, in control and bri1 mutants, to support the original findings of the Caesar et al. 2011 paper.

As shown by the phosphoproteomics data of Lin et al. (2015), the rapid phosphorylation of AHAs within 5 min after BR application occurs at Ser 315 and Thr 328 in the large cytoplasmic domain of the pumps and not at Thr 947. However, there is no anti-pSer315 or anti-Thr328 antibody available currently. Furthermore, the phosphorylation of AHAs at Thr 947 is not required for at least the BL-induced E_m_ hyperpolarization in tobacco leaf cells (Witthöft et al., 2011).

The representation of figures should be united to make the reading of the manuscript easier (such as figures 7CD and figure 8F and Supp.3).

We tried our best to do so.

The term nanocluster is used repeatedly, but the actual localization of the signaling components in nanoclusters is not tested in the paper. It might be worth imaging the nanocluster formation, or use the term nanocluster more carefully.

The reviewer is right. We changed the term “nanocluster” to “nano-scale organized BRI1 complex”.

It is not described in the methods how the quantification of the fluorophore numbers per PM area was done. Which membranes were measured and how? It is not clear which of the epidermal cell types were used for quantification – trichoblasts or atrichoblasts? I would also appreciate example images of the gradient distribution of AHA2 along the root, not only images from the meristem.

A more detailed description of the quantification can now be found under "Confocal imaging" in "Experimental Methods". A representative example for a confocal image of AHA2-GFP accumulation is given in Appendix 1 Figure 2.

In my opinion, standard deviation should be used to represent the error bars should be used instead of standard error of the mean.I would strongly recommend replacing the barcharts by scatterplots or boxplots.

SE has been changed to SD where statistically meaningful.

We did not switch to boxplot representations, as it makes the figures even more complex – especially those in which modelling and experimental data are combined. However, all original datasets are available so that the data representation can be reproduced when using the statistical procedures, we have specified.

[Editors' note: further revisions were suggested prior to acceptance, as described below.]

The manuscript has been improved but there are some remaining issues that need to be addressed, as outlined below:Experimental1) Link between BR and AHA: Manuscript by Li et al. 2022 in Plant Phys (https://academic.oup.com/plphys/advance-article/doi/10.1093/plphys/kiac194/6576642) should be discussed in the revised version as it carried data relevant to conclusions provided in this manuscript.

We discuss the results of this work in our paper. However, the measurements of in vitro AHA activity and AHA phosphorylation at Thr947 was performed after 2 h at the earliest.

It should also be mentioned that Li and colleagues (2022) used very high concentration of BL (1 µM) for seedling treatment. At least with respect to root growth this concentration appears to us to being beyond the physiological relevant concentration (Vukasinovic et al., 2021; our data).

The responses to the other issues of the revision are included in the “Response to the reviewers”.

Reviewer #2 (Recommendations for the authors):I appreciate the authors' efforts to address the criticisms. I believe the results are of interest. The model is better defined now and the flow of the manuscript has improved. There are more comparisons between simulations and experiments, as requested. Yet, the authors have not addressed entirely satisfactorily in my opinion the concerns raised. Here below I detail those aspects that remain unclear to me and which I suggest taking into account:1) (Related to Ad 2 # reviewer 2) My first issue comes from the conclusions they raise based on their experimental data from Figures 4 and 6, i.e. on the comparison between the response to BL of EZ and MZ cells. The authors have fully addressed my issue regarding this comparison: they have statistically assessed whether there are differences between MZ and EZ cells. They find that experimental data do not confirm a difference between MZ and EZ cells in pH value nor on its response to BL, as opposed to the model, which predicts that there is a difference. Specifically, the authors now state, lines 287-292: "However, due to the limitation in the sensitivity of the HPTS method and the biological variability in the different root preparations, the difference in the BL-induced acidification responses between MZ and EZ epidermal cells could not be statistically established. The modeling approach is therefore of great advantage for the detection and characterization of small, experimentally hardly comprehensible, cell physiological response differences (Appendix 1Figure 5)." I do not agree entirely with the last sentence. In my opinion there are, at least, two different options: Option 1) The modeling approach shows small differences which experimentally are difficult to be observed, as the authors claim in their last sentence; Option 2) Plants really do not have the differences predicted by the model. Hence, the model prediction (and hence the model) is not correct. I think Option 2 should not be omitted and hence I propose the last sentence is modified such that emphasis is made on the prediction that the model makes as opposed to the emphasis on assuming that the model is correct (and hence detects small differences).

We modified this sentence according to the reviewer´s suggestion. However we assume in this context, that the biological variety of the samples (e.g. physiological plasticity of root cells coming from different seedlings even if they were grown on the same plates and even if the seedlings are of the identical genotype) in combination with the technical variety with respect to sample preparation and handling will not make significant distinctiveness of the experimental data possible in principle. We see a major advantage to overcome these principle experimental limitations in the application of computational modeling.

2) (Related to AD. 4 from # reviewer 2) BRI1 requirement only at MZ (Pavelescu et al. Mol Sys Biol 2018) is not discussed.

The statement of Pavelescu and colleagues (2018) that BRI1 signaling in the meristematic zone is sufficient for root (elongation) growth is under debate. The work of Vukasinovic and colleagues (2021) show that BR biosynthesis is largely restricted to the root elongation zone (high BR concentration), where it overlaps with BR/BRI1 signaling maxima, and is low (low BR concentration) in the meristematic zone. This pattern eventually achieves optimal root organ growth in the long-term. Conversely, this means that functionally relevant BR/BRI1 perception and BR/BRI1 signaling occurs in the elongation zone. Otherwise, the establishment of such a BR gradient in combination with almost constant presence of BRI1 as well as of BAK1 and BIR3 in the epidermal cells of MZ and EZ would make no sense.

3) (Related to AD. 6 from # reviewer 2) In figure 7D the authors present simulation results for pH response under BL treatment for the WT and for bri301 in order to confirm the experimental data in Figure 7A-C. However, in 7A-C no pH measurements are made. Instead, the ion fluxes are measured. Can the simulation data be related more tightly to the experimental data? E.g. provide the ion fluxes, besides pH? Alternatively, measure pH in the plants?

We actually provided the pH-change comparison of wild type and *bri1-301* mutant as relative change of the 458/405 ratio in Figure 7B (treatment: 10 nM BL, 30 min). The results of the experimental endpoint determination (Figure 7B) agree well with modelling data of Figure 7D (10 nM BL, 20 min) and, moreover, with the H^+^ flux data.

4) (Related to AD. 8 and AD.10 from # reviewer 2) It still remains unclear how "cell wall instability" and "cell wall thickness" are computed. I think Equation for "cell wall thickness" in line 895 is incorrect (I think it is correctly computed in the manuscript but wrongly written down in that line). Besides, what is "Cell wall(t)" in the Eq. for "Cell wall instability" (line 896)?

Thank you for pointing out the error in the equation for “cell wall thickness”, which we have now corrected. We have further expanded the explanation in the supporting information. Briefly, the cell wall instability is induced by the acidification of the cell wall but limited by a maximal cell wall thickness based on experimental measurements by van Esse et al. (2011).

Reviewer #3 (Recommendations for the authors):In the re-submitted version of the manuscript and the response letter, the authors have solved most of my comments by discussing my concerns or by refusing them as being out of scope of the manuscript.I would like to point out several of these comments and the author's response:1) Strengthening the link between BR signaling and AHA activation:The authors argue that in the Lin et al., 2015 paper, phosphorylation of AHA at the residue Ser 315 and Thr 328 was demonstrated. It is, however, not clear whether this particular phosphorylation leads to AHA activation. Moreover, the Lin paper uses suspension cell culture, and this phosphorylation is apparent in the first 5 minutes, and in 3h of treatment it diminishes. Therefore my critique from the previous version remains – the central component of the model (activation of AHAs by BL signaling) should be experimentally verified.I understand that this is a difficult task. Possibly a proxy of the AHA activation could be used instead. For example, a simple root growth rate measurement experiment where BL would be added during the timecourse could help strengthen the rapid effect of BL on AHA activation, one would expect a similar effect as in case of fussicoccin – if BL activates AHAs, this should lead at least to a transient increase in growth rate. Such a simple readout of the AHA activity could be then used for wild type roots as well as the bri1-301 and cngc10 mutant.

As discussed in the previous rebuttal letter, the actual root growth at organ level is a longterm effect of BL action, which we do not address in this study.

However, we previously published two proxies of BL-induced AHA activation linked to apoplast acidification and elongation of root and hypocotyl cells (Caesar et al., 2011): PM hyperpolarization and wall swelling. At that time, we were not yet able to measure the apoplastic pH at cellular resolution. I would like to summarize the data here, which can be found in Figure 5 and Figure 6 of Caesar et al. (2011):

Short-time treatment of the cells with 50 µM *ortho*-vanadate (*o*V), an inhibitor of AHA activity, entirely blocks 10 nM BL-induced wall swelling and PM hyperpolarization.This block is reversible: After *o*V was washed out, the addition of 10 nM BL again induces wall swelling and PM hyperpolarization within 15 min.Addition of 2 µM fusicoccin (Fc) induces cell wall swelling and PM hyperpolarization in the absence of external BL. Addition of 10 nM BL to the Fc treated cells does only weakly enhance both responses, indicating that the system is almost saturated. I also included Author response image 1 that shows that the addition of 5 µM Fc induces the acidification of the cell wall in the elongation zone to similar extent than addition of 10 nM BL. We think that these proxies provide sufficient prove that the BL-responses are clearly linked to the AHA activity.

**Author response image 1. sa2fig1:** Ratiometric HPTS (458/405 emission ratio pH-dependent). Measurement of apoplastic pH in the root HPTS-8-hydroxypyrene-1,3,6-trisulfonic acid (trisodium salt).

In our opinion, these proxy data strongly suggest that the BL-induced cellular apoplastic pH change (or the ratio in the 458/405 emission spectra, respectively) is also due to regulation of AHA activity.

The reviewer is right that Lin and colleagues (2015) used suspension culture cells. However, the growth condition characteristics of the suspension cells are similar to those of root cells. In fact, the phosphorylation of Ser315 and Thr328 has not yet been proven to lead to a change in AHA activity. However, this prove is outside the scope of our manuscript. What is important for us here is the fact that a BR-induced phosphorylation of AHA is observed that correlates well in time with the initiation of the AHA-mediated cell physiological processes described here. The fact, that the phosphorylation of Ser315 and Thr328 decreases again 3 h after BR application, is expectable for a rapid and transient regulation of AHA activity in our opinion. Moreover, this observation would support our hypothesis that the further maintenance of AHA activity by BR is mediated by the phosphorylation of the penultimate threonine, which is reported to be detectable much later in time (≥ 2 h).

2) Localization of CNCG10 expressionI understand that cloning the CNGC10 promoter and translational fusions might be very laborious or out of scope of this manuscript. However, the authors claim in the text that the CNGC10 is expressed in all cell types of the root (according to eFP browser), and in the response to reviewers claim that CNGC10 is induced by protoplasting, and therefore the single cell analysis cannot be performed. However, the data in the eFP browser partially originate from protoplasted cells of the root, and in the eFP browser (http://bar.utoronto.ca/efp/cgi-bin/efpWeb.cgi ), I do not see protoplasting alert. Instead, CNGC10 seems to express predominantly in inner tissues of the root and is absent from epidermis (according to the 'root' and 'single cell' options). These expression data are only a proxy of the actual protein presence, however, I think the authors should discuss the predicted absence of CNGC10 from the root epidermis in the text.

We think this is a misunderstanding: The eFP browser data and the scRNAseq data show that *CNGC10* transcript is present in all tissues including the epidermal cells of the meristematic zone and elongation zone. These data are supported by the observation of Jin and colleagues (2015, 10.1007/s10265-014-0679-2, cited in the paper) that “*GUS analysis of a root cross section indicated that CNGC10 was expressed mainly in the endodermis and epidermis*”. The *CNGC10* transcript amount increases in the vascular tissue of the maturation zone, which is outside of the area of our interest. Of course, the eFP and scRNAseq data rely on the protoplasting of root cells. However, in contrast to, for instance, *BRI1, BIR3* and *AHA2*, protoplasting affects the *CNGC10* transcript amount according to the published scRNAseq data (Ma et al., 2020). Therefore, we cannot perform *CNGC10* pseudotime (temporal) analysis on the scRNAseq data, i.e., we cannot make conclusions about how the *CNGC10* transcript amount changes along the axis of the root tip. Actually, this is also found for *BAK1* transcripts, for which we also see a protoplasting effect in the scRNAseq data but not in the eFP browser presentation (no proto-alert in the root zones of our interest) Why there is no “proto-alert” in the eFP browser representation of *CNGC10* and *BAK1*, this remains an open question. However, the scRNAseq data are more crucial for us as they allow the pseudotime (temporal) analysis of crucial components (*BRI1*, *BRI3*, *AHA2*) so that they can be linked to the protein amount. The quantification of BAK1GFP is published, so that we were able to use this protein data (van Esse et al., 2011).

The reviewer is right that the *CNGC10* transcript amount appears low in the root tip.

However, transcript data are only a proxy for the CNGC10 protein amount and even less for CNGC10 activity. The production of transgenic *Arabidopsis* lines accumulating CNGC10GFP under the control of the native promoter is out of the scope of the manuscript – at least in our view.

3) Quantification of the AHA molecules along the longitudinal root axis.In the previous review, I asked the authors for more details about AHA quantification and requested some sample images of the AHA gradient. In the current version, the Appendix 1 Figure 2 shows several roots which look very stressed and show signs of dead cells in the meristematic zone, and have a short meristem and elongation zones. These roots do not look like healthy 7-d old roots. The example picture of AHA2-GFP does not show a continuous epidermal cell file on which the number of AHAs could be quantified. The figure legend states these are 7d old roots, while in the methods, the authors state that 5d old roots were used. I am worried that if such images were used for the quantification of AHA molecule numbers, the data might be noisy due to the status of these roots.

We can reassure the reviewer that we did not use such roots for quantification of GFP fusion proteins. For parallel evaluation of the fluorescence data by independent people, the roots were virtually divided into smaller sections. However, we wanted to show the reviewers entire roots in one and the same image. Unfortunately, we had problems with seedling/root germination and growth at the time of the first revision. We now present better root images of 5-days-old roots.

Moreover, I asked for a more detailed description of the method of molecule number quantification, the authors responded that now the method is described in the methods under "confocal imaging", but I could not find any description of the method in this section. Even though this manuscript is mainly a computation one, I think the authors should clearly explain and demonstrate that the experimental inputs are solid.

We are very sorry but the more detailed description of the confocal imaging was accidentally not included in the revised version of the manuscript. The description was now added.